# Lactococcus lactis NCDO2118 exerts visceral antinociceptive properties in rat via GABA production in the gastro-intestinal tract

Valérie Laroute[1†], Catherine Beaufrand[2†], Pedro Gomes[1,2], Sébastien Nouaille[1], Valérie Tondereau[2], Marie-Line Daveran-Mingot[1], Vassilia Theodorou[2], Hélène Eutamene[2*‡], Muriel Mercier-Bonin[2‡], Muriel Cocaign-Bousquet[1*‡]

[1]Toulouse Biotechnology Institute (TBI), Université de Toulouse, CNRS, INRAE, INSA, Toulouse, France; [2]Toxalim (Research Centre in Food Toxicology), Université de Toulouse, INRAE, ENVT, INP-Purpan, UPS, Toulouse, France

*For correspondence:
helene.eutamene@inrae.fr (HE);
cocaign@insa-toulouse.fr (MC-B)

[†]These authors contributed equally to this work
[‡]These authors also contributed equally to this work

Competing interest: The authors declare that no competing interests exist.

**Abstract** Gut disorders associated to irritable bowel syndrome (IBS) are combined with anxiety and depression. Evidence suggests that microbially produced neuroactive molecules, like γ-amino-butyric acid (GABA), can modulate the gut-brain axis. Two natural strains of *Lactococcus lactis* and one mutant were characterized *in vitro* for their GABA production and tested *in vivo* in rat by oral gavage for their antinociceptive properties. *L. lactis* NCDO2118 significantly reduced visceral hyper-sensitivity induced by stress due to its glutamate decarboxylase (GAD) activity. *L. lactis* NCDO2727 with similar genes for GABA metabolism but no detectable GAD activity had no *in vivo* effect, as well as the NCDO2118 ΔgadB mutant. The antinociceptive effect observed for the NCDO2118 strain was mediated by the production of GABA in the gastro-intestinal tract and blocked by GABA_B receptor antagonist. Only minor changes in the faecal microbiota composition were observed after the *L. lactis* NCDO2118 treatment. These findings reveal the crucial role of the microbial GAD activity of *L. lactis* NCDO2118 to deliver GABA into the gastro-intestinal tract for exerting antinociceptive properties *in vivo* and open avenues for this GRAS (Generally Recognized As safe) bacterium in the management of visceral pain and anxious profile of IBS patients.

## Editor's evaluation

Gut microbes produce neuromodulators including the neurotransmitter γ-aminobutyric acid (GABA), which could regulate pain and other neurological outcomes. This study examines how oral administration of the probiotic Lactococcus lactis affects visceral pain in rats, finding that a specific strain of L. lactis (NCD02118) suppresses stress induced pain in a manner dependent on bacterial GadB, the enzyme mediates GABA synthesis.

## Introduction

As defined by the Rome IV criteria, irritable bowel syndrome (IBS) is a functional gastro-intestinal disorder affecting 3–5% of adults in industrial countries. IBS results in various symptoms that strongly affect the patients' quality of life (*Dean et al., 2005*) and a significant socioeconomic burden (*Peery et al., 2012*). Abdominal pain is a prevalent IBS symptom, associated with changes in bowel habits (diarrhoea and/or constipation; *Camilleri et al., 2012*). Even though IBS pathophysiology remains not completely understood, IBS symptoms may originate from peripheral and/or central mechanisms

resulting in a dysfunctional gut-brain axis (*Fichna and Storr, 2012*). Anxiety and stressful psychological life events lead to central nervous system (CNS) disorder triggering in turn gut motility dysfunctions (*Mönnikes et al., 2001*) and strengthen visceral sensitivity (*Greenwood-Van Meerveld et al., 2016*). For a better understanding of the IBS pathophysiology, stress animal models have been developed mimicking IBS features, such as changes in visceral sensitivity and gut transit time (*Gué et al., 1997*).

Related to the complex pathophysiology of IBS and the diversity of IBS patients' profiles, the existing therapeutic strategies promote solutions usually limited to treat symptoms (motor/sensory disturbances). In addition, in several cases with psychological comorbidity, psychopharmacological drugs are used (*Dekel et al., 2013*). As an effective treatment, alternative probiotics were used to modulate visceral pain in IBS patients (*Moayyedi et al., 2010*; *Hungin et al., 2013*; *Ford et al., 2014*; *Didari et al., 2015*). However, no consensus results were obtained from human clinical studies (*Mazurak et al., 2015*) and mechanisms by which microbes signal from the gut lumen to the CNS, thus influencing pain perception, remain to be elucidated.

In recent years, mounting evidence has suggested that microbially produced neuroactive molecules can modulate the gut-brain axis communication (*Cryan and O'Mahony, 2011*; *Lyte, 2011*; *Reid, 2011*; *Cryan and Dinan, 2012*). For example, ingestion of a *Lactobacillus* strain, i.e., *L. rhamnosus* (*JB-1*) is able to regulate emotional behaviour and central γ-aminobutyric acid (GABA)ergic system in mouse via vagus nerve pathway (*Bravo et al., 2011*). Recently, GABA-modulating bacteria of the human gut microbiota have been associated to brain signatures related to depression (*Strandwitz et al., 2019*). In fact, GABA is the main inhibitory CNS neurotransmitter in mammals (*Wong et al., 2003*), and several important physiological functions have been characterized, such as neurotransmission, relaxing, and tranquilizer effects (*Hayakawa et al., 2007*; *Li and Cao, 2010*). GABA exerts these major functional effects *via* two GABA receptor subtypes, i.e., $GABA_A$ and $GABA_B$ (*Hyland and Cryan, 2010*) doing these receptors important pharmacological targets for clinically relevant anti-anxiety agents (*Foster and Kemp, 2006*).

In the human diet, glutamate constitutes up to 8–10% of amino acids; it is involved in gut protein metabolism and is the precursor of different important molecules produced within the intestinal mucosa, like GABA. GABA is synthetized by a pyridoxal-5'-phosphate (PLP)-dependent enzyme glutamate decarboxylase (GAD; EC 4.1.1.15) by irreversible α-decarboxylation of L-glutamate and consumption of one cytoplasmic proton (*Ueno, 2000*). Prokaryotic and eukaryotic cells are able to synthesize GABA, through the decarboxylation of glutamate and genes encoding GAD are present in the gut microbiota (*Mazzoli and Pessione, 2016*). Glutamate import and GABA export generally occur in bacteria simultaneously via a specific glutamate/γ-amino butyrate antiporter (*Ma et al., 2012*). The glutamate-dependent system is associated with acid resistance in many bacteria (*Feehily and Karatzas, 2013*).

Due to this large repertoire of beneficial effects, GABA has been classified as a health-promoting bioactive component in foods and pharmaceuticals (*Li and Cao, 2010*). Among bacteria, lactic acid bacteria (LAB) are generally recognized as safe sources to produce GABA at high levels and in an eco-friendly way (*Dhakal et al., 2012*). Numerous studies revealed that food-derived *lactobacilli* and gut-derived *Lactobacillus* and *Bifidobacterium* species are able to produce GABA *in vitro* (*Li and Cao, 2010*; *Yunes et al., 2016*). Interestingly, *Linares et al., 2016* reported the use of a novel *Streptococcus thermophilus* strain for the production of a naturally GABA-enriched yogurt. However, the link between bacterial GABA production *in vitro* and neuromodulatory activity *in vivo* remains poorly investigated. To date, only the human gut commensal GABA-producing *Bifidobacterium dentium* was shown *in vivo* to modulate sensory neuron activity in a rat faecal retention model of visceral hypersensitivity (*Pokusaeva et al., 2017*). Here, we focused on the transient food-borne LAB *Lactococcus lactis*. Frequently encountered in dairy products, *L. lactis* is one of the most ingested bacteria (*Mills et al., 2010*; *Laroute et al., 2017*). Although not considered as a commensal bacterium, *L. lactis* was found to persist transiently in the gut, depending on the strain under study (*Wang et al., 2011*; *Radziwill-Bienkowska et al., 2016*; *Zhang et al., 2016*). It was also demonstrated that *L. lactis* may exert *in vivo* a potent anti-inflammatory activity attenuating colitis (*Nishitani et al., 2009*; *Luerce et al., 2014*; *Ballal et al., 2015*; *Berlec et al., 2017*; *Nomura et al., 1999*). *L. lactis* is also able to produce GABA *in vitro* (*Nomura et al., 1999*) and the well-characterized NCDO2118 strain besides its probiotic properties (*Cordeiro et al., 2021*), is considered as an efficient GABA producer among

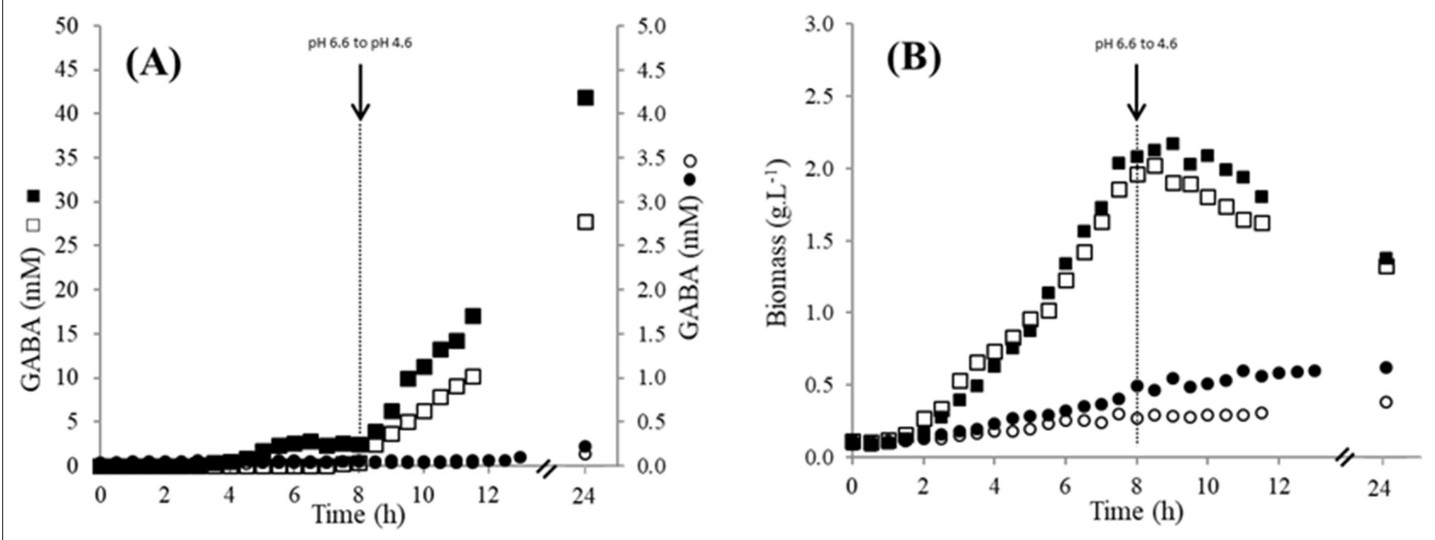

**Figure 1.** Growth and GABA production of the different *Lactococcus lactis* strains. (**A**) γ-Aminobutyric acid (GABA) production (mM); (**B**) evolution of biomass (g/L) during growth of *Lactococcus lactis* NCDO2118 (□, ■) or NCDO2727 (O, ●) in M17 supplemented with glutamate (8 g/L), arginine (5 g/L), glucose (45 g/L), and NaCl (300 mM). Two independent duplicates were performed.

The online version of this article includes the following source data for figure 1:

**Source data 1.** Table of biomass and GABA concentrations for *L. lactis* NCDO2118 and NCDO2727 strains.

the species (*Oliveira et al., 2017*; *Mazzoli et al., 2010*; *Oliveira et al., 2014*; *Laroute et al., 2016*; *Laroute et al., 2021*).

Here, we determined whether *L. lactis* NCDO2118 is able to deliver GABA *in vivo* via glutamate decarboxylation (GAD activity) and to exert GABAergic signalling-dependent antinociceptive properties in a stress-induced visceral hypersensitivity rat model.

## Results

### High in vitro GABA production by *L. lactis* NCDO2118 is related to its GAD activity

The NCDO2118 strain previously demonstrated to produce GABA in chemically defined medium (*Laroute et al., 2016*), was grown in batch bioreactor with complex medium allowing high production of biomass and GABA (see Experimental section). The biomass and GABA concentrations were measured all along the culture (*Figure 1A and B*). In these conditions, the maximal concentrations of GABA and biomass reached, respectively, 40 mM at 24 hr and around 2.0 g/L at 8 hr, which are higher than the previously reported values in chemically defined medium (*Laroute et al., 2016*).

The NCDO2727 strain was demonstrated here to have the *gad* operon involved in GABA biosynthesis similar to NCDO2118, in terms of genetic organization (gene sequences and promoter). *gadR*, *gadC*, and *gadB*, the three genes of the *gad* operon are located between *kefA* and *rnhB* like in the NCDO2118 strain and the sequences of the corresponding proteins, GadR, C, and B, are highly similar with more than 99% of identity. Only two single nucleotide polymorphisms and a deletion of one bp were identified in the *gadR* and *gadCB* promoters, respectively, compared to NCDO2118. However, when tested for its GABA-producing performances under similar culture conditions, GABA production was very weak (GABA concentration <0.2 mM, *Figure 1*). The biomass production was lower (maximal biomass around 0.5 g/L at 24 hr) than for the NCDO2118 strain (*Figure 1*). The GABA concentration did not increase when the biomass production increased during growth of NCDO2727 strain in medium with yeast extract and glutamate (data not shown). For *in vivo* experiments, the bacterial amount to be orally administrated was adjusted to treat animals with similar viable bacterial cell number for the two strains (i.e. $10^9$ CFU (Colony-Forming Units) per day; see below).

Interestingly, a high intracellular GAD activity was obtained at 7 hr in the NCDO2118 strain (45.3 ± 4.7 μmol/min.mg) consistently with its high GABA production ability, while in the NCDO2727 strain,

no activity was detected. The difference of GAD activity between the two strains was confirmed at 24 hr of culture (*Supplementary file 1*). This comparison demonstrated that *in vitro* GABA production could not be solely associated to GABA genetic determinants in *L. lactis*.

## Only GABA-producing *L. lactis* NCDO2118 prevents visceral hypersensitivity induced by stress in response to colorectal distension

The influence of a 10-day chronic treatment by *L. lactis* NCDO2118 or NCDO2727 on stress-induced visceral hypersensitivity to colorectal distension (CRD) is shown in *Figure 2*. Rats were given *L. lactis* strains by oral gavage once daily with washed bacterial cells at the same amount ($10^9$ CFU per day) whatever the strain under study. We first verified that GABA-producing *L. lactis* NCDO2118 in presence of glutamate had no impact on basal CRD sensitivity (*Figure 2—figure supplement 1*). Then, in vehicle-treated rats, a 2 hr of partial restraint stress (PRS) significantly increased the number of abdominal contractions compared to basal conditions for all the pressures of distention applied from 30 mmHg ($p < 0.001$, *Figure 2*). In presence of glutamate, oral administration of *L. lactis* NCDO2118 suppressed the PRS-induced enhancement of abdominal contractions ($p < 0.01$ for all CRD pressures applied, *Figure 2A*), restoring a quasi-basal sensitivity to CRD. However, in absence of glutamate, the NCDO2118 strain did not exert antinociceptive properties (*Figure 2A*). Glutamate alone, at the concentration used (0.2% [w/v]), had no impact on basal CRD sensitivity and on visceral hypersensitivity response (*Figure 2—figure supplement 2*). Oral administration of the low GABA-producing *L. lactis* NCDO2727 in presence of glutamate failed to reduce stress-induced visceral hypersensitivity (*Figure 2B*).

## The antinociceptive effect of *L. lactis* NCDO2118 is due to its ability to deliver GABA in vivo and is mediated by the activation of GABA_B receptors

The growth of the NCDO2118 $\Delta gadB$ mutant strain, in which GABA pathway was interrupted (see Experimental section for strain construction details), was close to that of the NCDO2118 strain under same culture conditions (*Figure 3—figure supplement 1*). However, this strain did not produce GABA (GABA concentration in the bioreactor <0.3 mM *Figure 3—figure supplement 1*) and had no detectable GAD activity. There was no impact of this strain on the visceral hypersensitivity response induced by stress in response to CRD in presence of glutamate (*Figure 3A*, $p < 0.05$ NCDO2118 $\Delta gadB$- vs. NCDO2118-treated animals). The beneficial effect of *L. lactis* NCDO2118 treatment on stress-induced visceral hypersensitivity ($P < 0.05$) was completely abolished when animals received the GABA_B receptor antagonist SCH-50911 in presence of glutamate (*Figure 3B*).

## GABA is produced by *L. lactis* NCDO2118 under 'stomach-like' conditions in vitro and in vivo in the stomach

Experiments were performed *in vivo* with rats fed *L. lactis* NCDO2118, vehicle or *L. lactis* NCDO2727, in the presence of glutamate for 10 days. For the NCDO2118 strain, the GABA level was high in the stomach and then tended to decrease in the other gut compartments ($p < 0.05$ in caecum vs stomach; *Figure 4A*). In contrast, in vehicle- or NCDO2727-treated animals, the GABA levels were similar all along the gastro-intestinal tract (*Figure 4B*). Consistently, in the stomach, the GABA production by *L. lactis* NCDO2118 was higher than that measured in vehicle or NCDO2727 strain (*Figure 4C*). To reinforce our findings on the potential role of the gastric region, we performed further experiments *in vitro* under 'stomach-like' conditions in the presence of glutamate (at pH = 4.6 in acetate buffer or gastric juice sampled from naive rats). In particular in rat gastric juice, we found that the rate of GABA production by the NCDO2727 strain was similar to the vehicle but the rate of GABA production by the NCDO2118 strain was fourfold increased (*Supplementary file 2*).

## Oral treatment with *L. lactis* NCDO2118 has minor influence on faecal microbiota abundance, composition, and diversity but influences specific genera

The composition of the faecal microbiota was not globally impacted by the *L. lactis* NCDO2118 treatment (*Figure 5*). Particularly, no changes in α and β diversities were observed (*Figure 5A and*

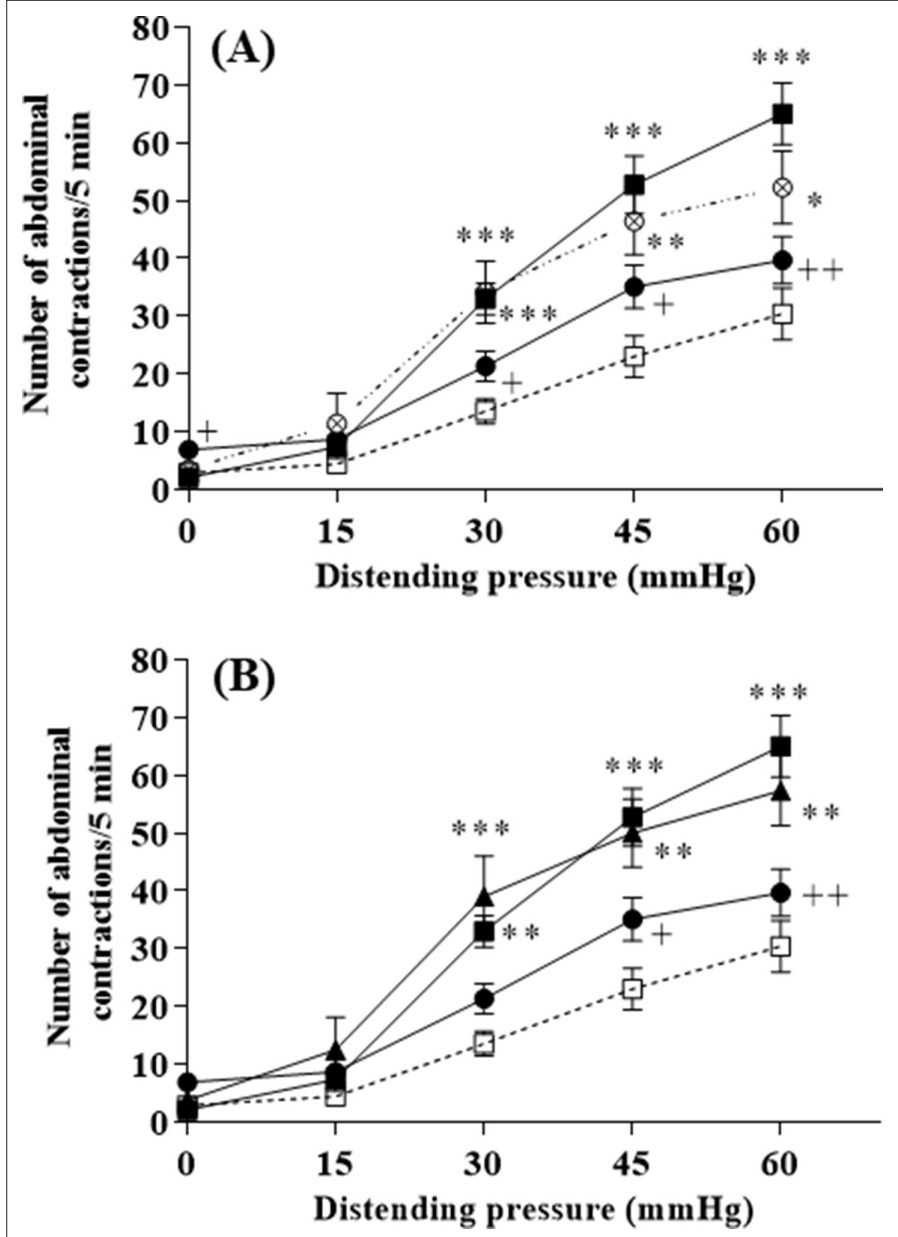

**Figure 2.** *L. lactis* effect on visceral hypersensitivity induced by stress in response to colorectal distension. (**A**) Effect of 10-day oral administration of γ-aminobutyric acid (GABA)-producing *Lactococcus lactis* NCDO2118 ($10^9$ CFU per day) in presence or absence of glutamate on PRS-induced visceral hypersensitivity at all the distension pressures of colorectal distension (CRD; from 15 to 60 mmHg). Data are expressed as means ± SEM (n=9 for the 'vehicle' group [□] and 'PRS + vehicle' group [■]; n=7 for 'PRS + NCDO2118' group [⊗]; n=12 for the 'PRS + NCDO2118 + glutamate' group [●]); (**B**) Effect of 10-day oral administration of GABA-producing *L. lactis* NCDO2118 and low GABA-producing *L. lactis* NCDO2727 ($10^9$ CFU per day) in presence of glutamate on PRS-induced visceral hypersensitivity at all the distension pressures of CRD. Data are expressed as means ± SEM (n=9 for the 'vehicle' group [□] and 'PRS + vehicle' group [■]; n=12 for the 'PRS + NCDO2118 + glutamate' group [●]; n=8 for the 'PRS + NCDO2727 + glutamate' group [▲]). $^*p<0.05$, $^{**}p<0.01$, $^{***}p<0.001$ vs. basal values for animals treated with vehicle. $^+p<0.05$, $^{++}p<0.01$ vs. values for stressed animals treated with vehicle.

The online version of this article includes the following source data and figure supplement(s) for figure 2:

**Source data 1.** Effect of 10-day oral administration of GABA-producing *L. lactis* NCDO2118 on PRS-induced visceral hypersensitivity.

*Figure 2 continued on next page*

*Figure 2 continued*

**Source data 2.** Effect of 10-day oral administration of GABA-producing *L. lactis* NCDO2118 and low GABA-producing *L. lactis* NCDO2727 on PRS-induced visceral hypersensitivity.

**Source data 3.** Incidence of 10-day oral administration of *L. lactis* NCDO2118 in presence of glutamate on basal visceral sensitivity.

**Source data 4.** Incidence of 10-day oral administration of glutamate on basal visceral sensitivity and PRS-induced visceral hypersensitivity.

**Figure supplement 1.** Incidence of 10-day oral administration of *L. lactis* NCDO2118 in presence of glutamate on basal visceral sensitivity at all the distension pressures of CRD (from 15 to 60 mmHg).

**Figure supplement 2.** Incidence of 10-day oral administration of glutamate on basal visceral sensitivity and PRS-induced visceral hypersensitivity at all the distension pressures of CRD (from 15 to 60 mmHg).

---

*C*). Besides, the non-parametric Kruskal-Wallis test on each of the 150 identified genera present in samples revealed few differences in taxon abundances in faecal microbiota, in particular significantly for three genera, *Clostridium* (p=0.023), *Frisingicoccus,* and *Papillibacter* (p=0.041), and nearly significantly for *Escherichia* genus (p=0.069), after *L. lactis* NCDO2118 treatment (*Supplementary file 3*).

## Discussion

For the first time, we demonstrated that a transient food-borne LAB *L. lactis* (strain NCDO2118) exerts, in an animal model of acute stress, visceral anti-hypersensitivity effects via its ability to produce significant levels of GABA *in vivo*, due to an active GAD enzymatic potential and not solely to the presence of *gadB* gene. Indeed, the low GABA producer NCDO2727 strain and NCDO2118 Δ*gadB* mutant strain unable to produce GABA, fail to exert such antinociceptive effect. This effect is mediated by the metabolically active NCDO2118 which delivers GABA in the gastro-intestinal tract, that in turn influences host GABAergic host physiological system.

*L. lactis* is a very common bacterium widely used in food industry with a GRAS status. Bacteria can metabolize, via decarboxylation mechanisms, several amino acids to synthetize active amino compounds. In particular, GAD catalyses the irreversible α-decarboxylation of glutamate into GABA. Even though genes encoding GAD are found in numerous bacteria present in different environments and ecosystems, i.e., native soil, natural landscape as well as in the gastro-intestinal tract, the ability of these bacteria to produce GABA has not systematically been evidenced. Interestingly, among *L. lactis* bacteria studied here, NCDO2727 was qualified as a low producer of GABA and NCDO2118 as a high producer. Although these two strains harbour similar GAD encoding *gadB* gene, they differed significantly at the level of their GAD activity. Because the promoters of *gad* genes are nearly identical in the two strains, transcriptional regulations should not explain the observed differences. The growth defect of NCDO2727 should also not be involved since higher growth of the strain in another culture medium did not restore a high GABA production. Herein, we demonstrate *in vivo* that only a 10-day chronic oral administration of the high producer NCDO2118 strain displaying a high GAD activity is able to reduce visceral hypersensitivity induced by stress. This effect was accompanied by an increased GABA production in the gastric compartment. Taken together, all these observations suggest that transient NCDO2118 strain is metabolically active *in vivo* to deliver GABA in the gastro-intestinal tract lumen. We evaluated the impact of the NCDO2118 treatment on the faecal microbiota and observed no changes in α or β diversity. This result is consistent with observations in many other studies on IBS that have shown that probiotic treatment did not shift the overall diversity of the intestinal microbiota of mice, rats, or humans (*Cussotto et al., 2021*; *Grazul et al., 2016*; *Jeffery et al., 2020*; *Labus et al., 2017*; *Tap et al., 2017*). Indeed, in 'healthy microbiota' the addition of a probiotic does not promptly impact the resident microbial populations but rather could have a significant impact on reshaping the microbiota in an already existing dysbiosis humans (*Alander et al., 1999*; *Eloe-Fadrosh et al., 2015*; *Lee et al., 2004*). We next compared genera abundancies after treatment using non-parametric tests and found an enrichment of the *Clostridium* genus after *L. lactis* NCDO2118 treatment compared to the vehicle group. This genus was shown in silico as modulating GABA (i.e. GABA producer and consumer) in humans (*Strandwitz et al., 2019*). An enrichment of the *Escherichia* genus was also observed. Interestingly, in the study on the *E. coli* Nissle 1917 strain, Pérez-Berezo and colleagues showed that GABA was not able to directly cross the epithelial barrier (*Pérez-Berezo et al., 2017*),

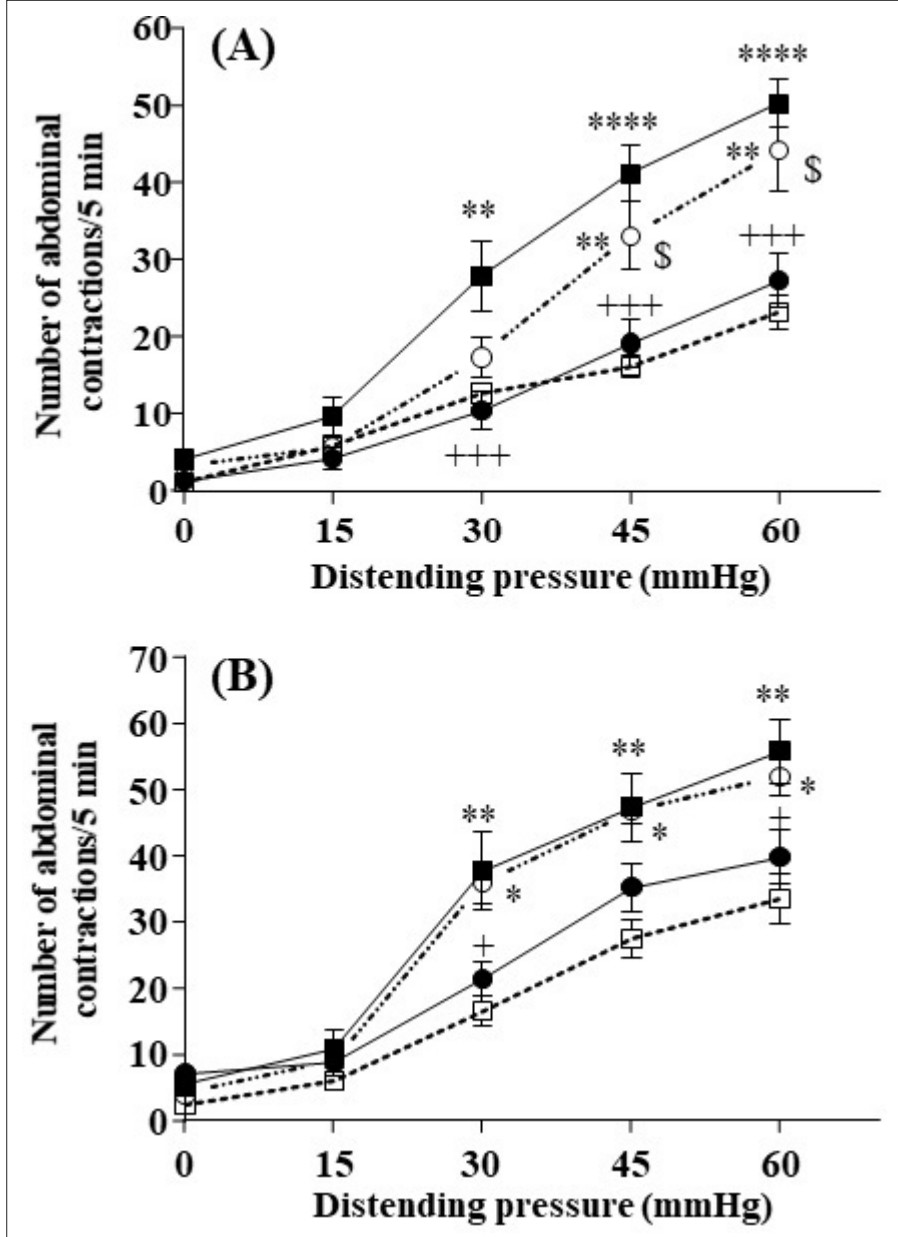

**Figure 3.** The anti-nociceptive effect of *L. lactis* NCDO2118. (**A**) Effect of 10-day oral administration of γ-aminobutyric acid (GABA)-producing *Lactococcus lactis* NCDO2118 and non-GABA producing NCDO2118 *ΔgadB* mutant ($10^9$ CFU per day) on PRS-induced visceral hypersensitivity at all the distension pressures of colorectal distension (CRD; from 15 to 60 mmHg). Data are expressed as means ± SEM (n=9 for the 'vehicle' group [□] and 'PRS + vehicle' group [■]; n=12 for the 'PRS + NCDO2118 + glutamate' group [●]; n=13 for the 'PRS + NCDO2118 *ΔgadB* + glutamate' group [○]); (**B**) Effect of 10-day oral administration of GABA-producing *L. lactis* NCDO2118 ($10^9$ CFU per day) on PRS-induced visceral hypersensitivity at all the distension pressures of CRD in presence or not of $GABA_B$ receptor antagonist SCH-50911 (3 mg/kg bw, IP). Data are expressed as means ± SEM (n=9 for the 'vehicle' group [□] and 'PRS + vehicle' group [■]; n=12 for the 'PRS + NCDO2118 + glutamate' group [●]; n=7 for the 'PRS + NCDO2118 + glutamate + SCH-50911' group [○]). *$p<0.05$, **$p<0.01$, ****$p<0.0001$ vs. basal values for animals treated with vehicle. +$p<0.05$, +++$p<0.001$ vs. values for stressed animals treated with vehicle. $p<0.05$ vs. for stressed animals treated with NCDO2118 + glutamate.

The online version of this article includes the following source data and figure supplement(s) for figure 3:

**Source data 1.** Effect of 10-day oral administration of GABA-producing *L. lactis* NCDO2118 and non-GABA producing NCDO2118 delta-gadB mutant on PRS-induced visceral hypersensitivity.

*Figure 3 continued on next page*

*Figure 3 continued*

**Source data 2.** Effect of 10-day oral administration of GABA-producing *L. lactis* NCDO2118 on PRS-induced visceral hypersensitivity in presence or not of GABAB receptor antagonist SCH-50911.

**Source data 3.** Table of biomass and GABA concentrations for *L. lactis* NCDO2118 and its ΔgadB mutant.

**Figure supplement 1.** Evolution of biomass (g/L) during growth of *Lactococcus lactis* NCDO2118 ΔgadB mutant (♦) and evolution of γ-aminobutyric acid (GABA) production (▲) compared to *L. lactis* NCDO2118 (biomass □ and GABA Δ) in M17 supplemented with glutamate (8 g/L), arginine (5 g/L), glucose (45 g/L), and NaCl (300 mM).

requiring the C12AsnGABAOH lipopeptide for functionalization and subsequent translocation. It is then possible that the enrichment of the genus *Escherichia* in the faecal microbiota of *L. lactis*-treated animals could help reduce visceral pain via a lipopeptide-mediated GABA functionalization pathway. Of note, in our study, using the same method as previously described (*Pérez-Berezo et al., 2017*), we were not able to detect any C12AsnGABAOH lipopeptide in *L. lactis* NCDO2118 (results not shown). Other yet unidentified lipopeptides could be at play, with the possible contribution of other bacteria capable of producing such functional GABA. Altogether, further investigations are needed for a better understanding of the site(s) of interaction between luminal GABA delivery by *L. lactis* NCDO2118 and the gastro-intestinal barrier and its environment.

GABA is the major inhibitory neuromediator, mainly involved in regulating physiological and psychological responses. Today, indirect evidences point out the effect of gut microbiota as well as probiotic strains (*Cryan and Dinan, 2012*) on CNS by modulating the GABAergic system. Moreover, alterations of this GABAergic system have a key role in the development of stress-related psychiatric disorders (*Schür et al., 2016*). Converging evidence, using genetic and pharmacological approaches, illustrates the important role of GABA$_B$ receptors in anxiety disorders. Using a specific antagonist (SCH-50911), we demonstrated the involvement of GABA$_B$ receptor in the visceral antinociceptive effect mediated by the delivery of GABA produced by the NCDO2118 strain. GABA$_B$ receptors are strongly expressed in the gastro-intestinal tract (*Hyland and Cryan, 2010*) and widely distributed from the stomach to the ileum in the enteric nervous system (ENS; *Auteri et al., 2015*). In rodents, previous studies established the presence of autocrine and paracrine GABA signalling mechanisms in gastro-intestinal epithelial cells via GABA receptor activation. In rat, GABA$_B$ receptors have also been identified on the mucosal gland cells of the stomach (*Castelli et al., 1999*; *Erdo and Bowery, 1986*; *Erdö et al., 1990*; *Krantis et al., 1994*). Actually, the GABA$_B$ metabotropic receptors in the gastro-intestinal tract are known to regulate several gut functions and gut-to-brain signalling pathway (*Hyland and Cryan, 2010*). In mice, selective stimulation of GABA$_B$ receptors increases gastric acid secretion both through vagal cholinergic and gastrin-dependent mechanisms (*Piqueras and Martinez, 2004*). These combined central and peripheral mechanisms result in an increase in the luminal acidity, which provides a supportive environment for the GAD activity of *L. lactis* NCDO2118. Accordingly, and despite a limited number of animals, we observed herein a significant level of GABA production measured after 10-day administration of *L. lactis* NCDO2118 in the stomach, which was supported by an *in vitro* GABA production in gastric juice sampled from naive rats and supplemented with glutamate; in this way, a dynamic 'virtuous circle' would be generated between *L. lactis* NCDO2118 and the host. Furthermore, vagal and splanchnic afferents of GABA$_B$ receptors are involved in the modulation of sensitivity (*Hyland and Cryan, 2010*). Probiotic strains have been described to modulate brain functions via a dependent vagus activation pathway (*Bravo et al., 2011*). This study was the first to demonstrate the ability of *Lactobacillus* to modify central levels of GABA through the stimulation of the vagus nerve. This central GABA release goes hand with beneficial effect on emotional behaviour impairments induced by stress. However, the primary molecular mechanisms underlying how *Lactobacillus* stimulated vagal afferents were not resolved. Here, we could firmly demonstrate that a neurotransmitter, i.e., GABA, directly produced by *L. lactis* NCDO2118 into the gastro-intestinal tract lumen, exerted beneficial effect on stress-induced gut visceral hypersensitivity via GABA$_B$ receptor stimulation. Even though we do not investigate the activation of the ENS or the CNS herein, our results associated with previous published data bring strengths on the table in the functional communication between bacteria, gut, and brain.

In conclusion, our data strongly suggest that, according to its ability to deliver into the gastro-intestinal tract lumen neurometabolites like GABA in significant levels, *L. lactis* NCDO2118 could be considered as a helpful candidate in the management of functional gastro-intestinal disorders

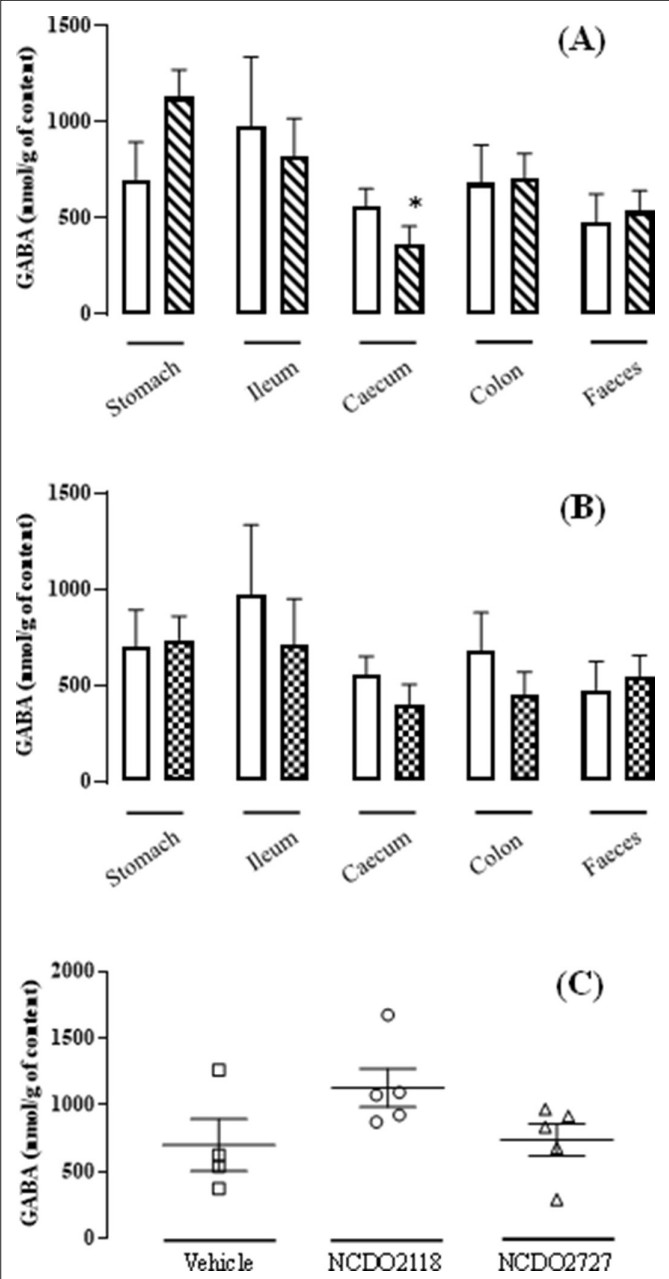

**Figure 4.** *L. lactis* effect on the GABA production all along the gastro-intestinal tract. (**A**) Variation of the γ-aminobutyric acid (GABA) concentration all along the gastro-intestinal tract for the group 'NCDO2118 + glutamate' (◨) vs. the group 'vehicle' (□) after a 10-day daily oral administration at $10^9$ CFU per day. (**B**) Variation of the GABA concentration all along the gastro-intestinal tract for the group 'NCDO22727 + glutamate' (▨) vs. group 'vehicle' (□) after a 10-day daily oral administration at $10^9$ CFU per day. (**C**) Variation of the GABA concentration in the gastric content for the group 'vehicle' (□), the group 'NCDO2118 + glutamate' (○), and the group 'NCDO2727 + glutamate' (△) after a 10-day daily oral administration at $10^9$ CFU per day. Data are expressed as means ± SEM. Non-parametric Kruskal-Wallis test, supplemented by Dunn's multiple comparison test, revealed no statistical differences between samples except caecum vs. stomach for the NCDO2118 strain (*$p < 0.05$).

The online version of this article includes the following source data for figure 4:

**Source data 1.** Variation of the GABA concentration all along the gastro-intestinal tract for NCDO2118+glutamate.

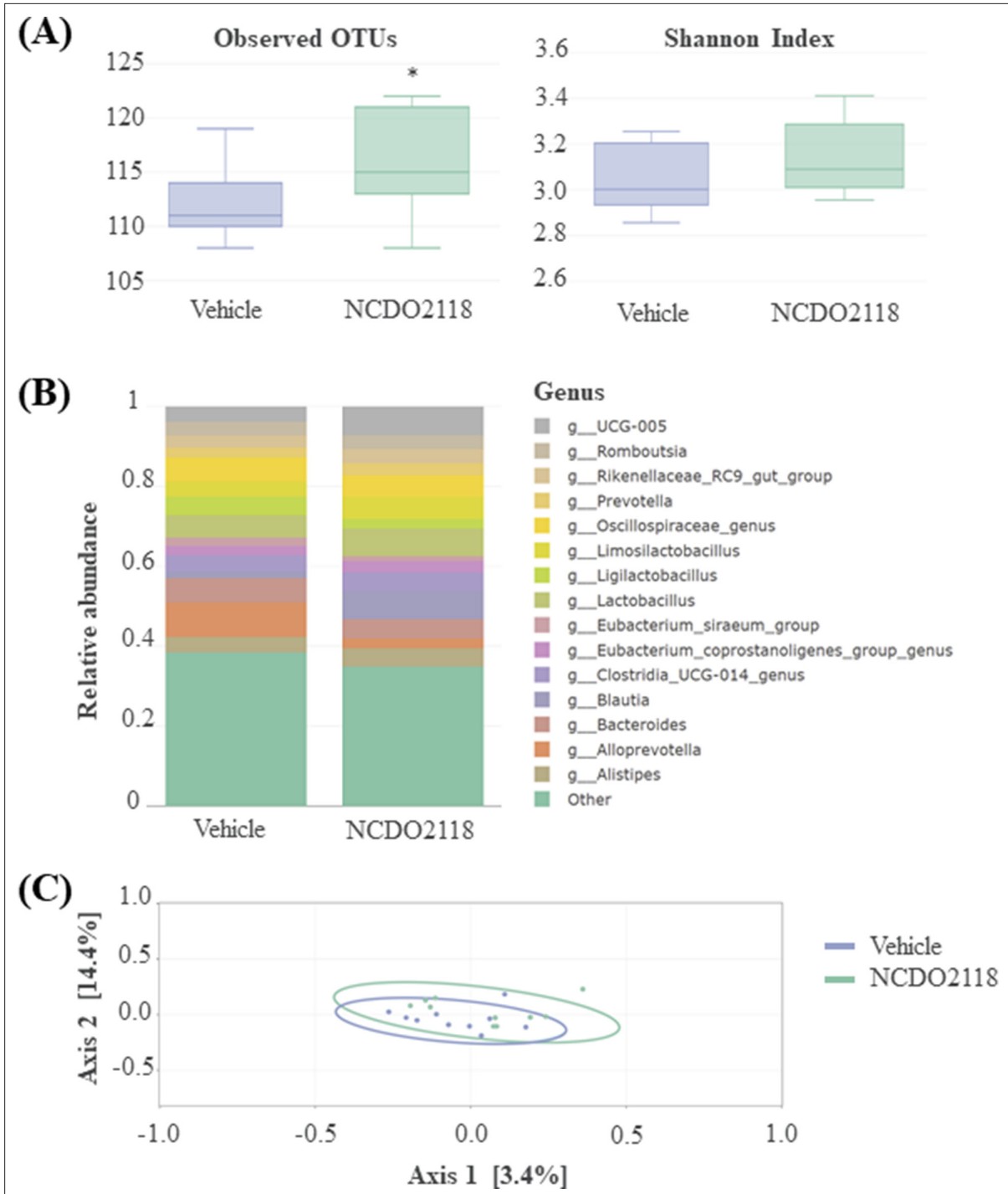

**Figure 5.** Overview of the faecal microbiota after a 10-day daily oral administration of γ-aminobutyric acid (GABA)-producing *L. lactis* NCDO2118 ($10^9$ CFU per day) in presence of glutamate vs. 'vehicle' group. (**A**) Alpha diversity within faecal samples. Two measures are shown: mean observed number of OTUs per sample, an estimate of richness (left panel), and Shannon Index, indicating the evenness of the sample (right sample). One-way ANOVA followed by the post hoc pairwise Tukey's test revealed no statistical differences between samples ($p > 0.05$). (**B**) Top 15 dominant bacterial genera in faecal samples; (**C**) MDS ordination plot of the Bray-Curtis distance between samples, a representation of phylogenetic similarity. PERMANOVA revealed no statistical differences between samples ($p > 0.05$).

associated with stressful situations. Regarding our results, GAD activity rather than genetic organization of *gad* genes appears as a key determinant for in vivo antinociceptive properties of *L. lactis*. Therefore, this work opens perspectives for GRAS *L. lactis* strains as future therapeutic agents for the management of visceral pain and the anxious profile of IBS patients.

## Materials and methods

### Bacterial strains, medium, and culture conditions

The strains NCDO2118 and NCDO2727, two *L. lactis* subsp. *lactis* from vegetable origin, were used throughout this study. *L. lactis* NCDO2118 Δ*gadB* mutant was constructed by double crossing over in the chromosome as previously described (*Maguin et al., 1996*). Briefly, two fragments upstream and downstream of the *gadB* coding sequence were PCR amplified, fused by overlapping PCR, and cloned in the pGhost9 vector using Gibson assembly method (New Englang Biolabs). Primers are listed in *Supplementary file 4*.

Bacterial cultures were performed in duplicate in 2 L Biostat B-plus bioreactor (Sartorius, Melsungen, Germany) in M17 supplemented with 55 mM (8 g/L) glutamate, 29 mM (5 g/L) arginine, 250 mM (45 g/L) glucose, and 300 mM NaCl. Cultures were incubated at 30°C. Fermentations were carried out under oxygen-limiting conditions. pH was maintained at 6.6 by KOH addition for 8 hr, then pH was dropped and regulated at 4.6. Culture was inoculated with cells from pre-cultures grown in Erlenmeyer flask on similar medium, harvested during the exponential phase, and concentrated in order to obtain an initial optical density at 580 nm of 0.25 in the fermenter. Bacterial growth was estimated by spectrophotometric measurements at 580 nm (Libra S11, Biochrom, BIOSERV, Massy, France; 1 Unity of absorbance is equivalent to 0.3 g dry weight/L). Samples were collected every 30 min for HPLC (High Performance Liquid Chromatography) measurement of GABA concentration in the growth medium. For *in vivo* assays and measurement of bacterial strain activity in the gastric juice *in vitro*, cells were harvested before the pH modification (i.e. at 7 hr for NCDO2118 and NCDO2727). The culture volume required for approximately $3 \times 10^{11}$ CFU (colony-forming units) was centrifuged to pellet the bacterial cells. Then, cells were washed and suspended in 0.9% (m/v) NaCl containing 15% glycerol (v/v) to a final concentration of $10^9$ CFU/mL. For GAD activity measurements, 150 mg of bacterial cells were harvested at 7 hr and 24 hr of culture.

### Sequence analysis of *gad* operon

In the NCDO2118 strain, the gad operon sequence was extracted from the chromosome sequence deposited in NCBI-GenBank database under the accession number CP009054. The *gad* operon in NCDO2727 strain was sequenced and deposited in NCBI-GenBank database under the accession number MK225577. Briefly, DNA was amplified using two primers, GadSeq_F (5' –TCCAGAAATAAC AGCTACATTGACATAATG –3') and GadSeq_R (5'– TAACAGCCCCATTATCTAAGATTACTCC –3') and the Q5 High-Fidelity DNA polymerase (NEB) and sequenced by Eurofins Genomics (primers listed in *Supplementary file 4*). Sequences of *gad* operon were compared by Blast alignment algorithm.

### Animals and surgical procedure

Adult female Wistar rats (200–225 g) were purchased from Janvier Labs (Le Genest St Isle, France) and individually housed in polypropylene cages under standard conditions (temperature 22 ± 2°C and a 12 hr light/dark cycle) with free access to water and food (standard pellets 2016, Envigo RMS SARL, Gannat, France). All experiments were approved by the Local Animal Care and Use Committee (APAFiS#5577–201606061639777 v3, see Colorectal and Under PRS conditions sections, and APAFiS#14898–2018043016031426, see Without PRS) in compliance with European directive 2010/63/UE.

### Colorectal distension procedure and acute stress procedure

Under general anaesthesia by intraperitoneal administration of 0.6 mg/kg acepromazine (calmivet, Vetoquinol, Lure, France) and 120 mg/kg ketamine (Imalgene 1000, Merial, Lyon, France), rats were equipped with NiCr wire electrodes implanted in the abdominal striated muscle for EMG recording (*Morteau et al., 1994*). Animals were then accustomed to be in polypropylene tunnels for several days before CRD. A 4-cm long latex balloon, fixed on rigid catheter was used. CRD was performed

after insertion of the balloon in the rectum at 1 cm from the anus. The tube was fixed at the basis of the tail. Isobaric distensions of the colon were performed from 0 to 60 mmHg using a Distender Series IIR Barostat (G&J Electronics Inc, Toronto, Canada) with each distension step lasting 5 min. The striated muscle spike bursts, related to abdominal cramps, were recorded on an electroencephalograph machine (Mini VIII, Alvar, Paris, France).

PRS, a relatively mild non-ulcerogenic model of stress, was performed as previously described (*Williams et al., 1988*). Briefly, rats were sedated with diethyl-ether and their fore shoulders, upper forelimbs and thoracic trunk were wrapped in a confining harness of paper tape to restrict, but not prevent, body movements. Rats were then placed in their home cage for 2 hr.

## Experimental protocol for in vivo assays
### Under PRS conditions
Series of experiments, based on a 10-day treatment by oral gavage (as detailed below), were conducted using, for each series, three groups of 7–12 female rats equipped for EMG. For all oral treatments used, basal and post-PRS (20 min after the 2 h PRS session) abdominal responses to CRD were recorded on day 9 and day 10, respectively.

In the first series of experiments, groups of rats were orally treated for 10 days (1 mL/rat) with *L. lactis* NCDO2118 ($10^9$ CFU/day) plus glutamate (0.2% [w/v]) or not, *L. lactis* NCDO2727 plus glutamate (0.2% [w/v]) or vehicle (NaCl 0.9% [w/v] + glycerol 15% [v/v]). In the second series of experiments, rats were divided into three groups and orally treated for 10 days (1 mL/rat) with *L. lactis* NCDO2118 ($10^9$ CFU/day) plus glutamate (0.2% [w/v]), *L. lactis* NCDO2118 plus glutamate (0.2% [w/v]), and GABA$_B$ receptor antagonist (2 S) (+)–5,5-dimethyl-2-morpholineacetic acid (SCH-50911, Sigma-Aldrich SML1040; 3 mg/kg body weight, IP 20 min before the PRS session) or vehicle (NaCl 0.9% [w/v] + glycerol 15% [v/v]). In the last series of experiments, rats were divided into three groups and orally treated for 10 days (1 mL/rat) with *L. lactis* NCDO2118 ($10^9$ CFU/day) plus glutamate (0.2% [w/v]), *L. lactis* NCDO2118 *ΔgadB* ($10^9$ CFU/day) plus glutamate (0.2% [w/v]) or vehicle (NaCl 0.9% [w/v] + glycerol 15% [v/v]).

### Without PRS
In order to determine GABA levels at different locations of the gastro-intestinal tract, further experiments without PRS application were performed as follows: three groups of rats (n=5 per group) were orally treated for 10 days (1 mL/rat) with *L. lactis* NCDO2118 ($10^9$ CFU/day) plus glutamate (0.2% [w/v]), *L. lactis* NCDO2727 plus glutamate (0.2% [w/v]), or vehicle (NaCl 0.9% [w/v] + glycerol 15% [v/v]). After the last administration, deep inhalation anaesthesia with Isoflurane, followed by terminal aortic blood sampling, was performed. After the sacrifice, gastric, ileal, caecal, and colonic contents, as well as faeces, were collected for GABA measurements (see GABA extraction and quantification).

## GAD activity and GABA production by *L. lactis* cells
For GAD activity measurement, 150 mg bacterial cells were washed twice with 0.2% KCl (w/v) and suspended in 3 mL sodium acetate buffer (100 mM, pH 4.6) containing 4.5 mM MgCl$_2$, 22% (v/v) glycerol, and 1.5 mM dithiothreitol. This mixture was distributed into three tubes containing 6 mg of glass beads. Then, cells were disrupted in a FastPrep-24 homogenizer (MP Biomedicals, Illkirch, France) using six cycles of 30 s at 6.5 m/s interrupted by 1 min incubation on ice. Cell debris was removed by centrifugation for 15 min at 10,000 *g* and 4°C. The supernatant was used for enzyme assays, and the protein concentration of the extract was determined by the Bradford method. Enzyme assay was realized with 0.5 mL of substrate solution, consisting of 20 mM sodium glutamate, 2 mM PLP incubated at 30°C then mixed with 0.5 mL supernatant. Every 30 min until 4 hr, 100 μL were sampled and inactivated by boiling for 5 min to stop the decarboxylation reaction. Reaction mixtures were subsequently analysed for the presence of GABA using HPLC.

The kinetics of GABA production by *L. lactis* NCDO2118 and *L. lactis* NCDO2727 was monitored *in vitro* at 37°C as follows: 0.5 mL of each strain ($10^9$ CFU/mL) or the NaCl/glycerol vehicle was added to 1 mL of 100 mM acetate buffer pH = 4.6 or gastric content from naive rat and 0.5 mL glutamate 0.4% (w/v). 100 μL of assay mixture were sampled at regular time intervals (until 60 min) and inactivated by boiling to stop the decarboxylation reaction. Reaction mixtures were subsequently analysed

using HPLC. The initial rate was determined as the amount glutamate converted into GABA (μmol) per minute.

## GABA extraction and quantification

GABA concentration in culture supernatant or in reaction mixtures, associated to assays for GAD activity and GABA production, was measured by HPLC (Agilent Technologies 1200 Series, Waldbronn, Germany) as previously described (*Laroute et al., 2016*).

GABA measurements were realised in samples from different digestive tract locations (gastric, ileal, caecal, and colonic contents and faeces). GABA was extracted from 200 mg of contents with methanol (three times with 3 mL) at room temperature. The mixture was centrifuged and the supernatant was transferred into tube and concentrated to dryness. Dried samples were resuspended in 100 μL methanol then analysed by HPLC as described above.

## Faecal microbiota composition using 16S rRNA gene sequencing

Faeces were collected at the end of the 10-day oral treatment, just before the 2 hr PRS session (see Under PRS conditions). Faecal samples were stored at –80°C until DNA was extracted using the ZymoBIOMICS DNA Miniprep Kit (D4300, Zymo Research) following manufacturer's instructions. The 314 F/805 R primers (5′ GACTACHVGGGTATCTAATCC-Forward primer), 5′ GACTACHVGGGTATCT AATCC-Reverse primer) were used to amplify the V3-V4 variable regions of the 16 S rRNA gene. Forward primer and reverse primer carried overhang adapters (5′ CTTTCCCTACACGACGCTCTTCCG ATCT-Forward primer, (5′ GGAGTTCAGACGTGTGCTCTTCCGATCT-Reverse primer) for Illumina index and sequencing adapters. First round of PCR was carried in a single-step 30 cycles using the MTP Taq DNA Polymerase Kit (D7442, Sigma-Aldrich) under the following conditions: 94°C for 1 min, followed by 30 cycles of 94°C for 1 min, 65°C for 1 min, and 72°C for 1 min, after which a final elongation step at 72°C for 10 min was performed. Second-round amplicons libraries and sequencing were performed at the Sequencing Platform of Toulouse (GeT-Biopuces) on an Illumina-MiSeq following the manufacturer's guidelines.

Bioinformatic treatments of all samples were performed with the Find Rapidly OTUs with Galaxy Solution pipeline (*Escudié et al., 2018*). Briefly, reads were contiaued with the VSEARCH software and sequences with sizes inferior to 300 bp or superior to 700 bp were eliminated. A clustering at 97% similarity was used to define OTUs with the SWARM algorithm and chimeric sequences were also removed. Sequences were then filtered using the phiX contaminant databank and the OTUs presenting a frequency inferior to 0.005% on all samples and present in less than three samples were also removed (*Bokulich et al., 2013*). The R package ANOMALY (*Fisch et al., 2022*) was used for the assignment of the taxonomic classification of the representative sequences of each OTU. For that, the algorithm IDTAXA and the reference databases SILVA 138 16 S and GTDB bac120_arc22 were used (*Parks et al., 2020*; *Quast et al., 2013*). Alpha (α) and beta (β) diversities and differential abundancies analyses were performed in R using the Phyloseq package (*McMurdie et al., 2013*). Alpha diversity metrics (i.e. observed OTUs and Shannon Index) were calculated based on the genus level. One-way ANOVA was used to test α diversity metric dissimilarities with post hoc Tukey's test. The Bray-Curtis distance was used to measure β diversity metrics. We explored the community structure of the samples with PERMANOVA (*Kelly et al., 2015*).

## Statistical methods

The software GraphPad Prism 9.1 (GraphPad, San Diego, CA) was used for other statistical analyses. For animal experiments, data are reported as the means ± SEM. One-way ANOVA, followed by Tukey's Multiple Comparison test, was performed to compare data between the different groups of animals. For GABA measurements, data are reported as the means ± SEM and the non-parametric Kruskal-Wallis test, supplemented by Dunn's multiple comparison test, was used. Statistical significance was accepted at $p < 0.05$.

## Acknowledgements

This work received the financial support of Toulouse-Tech Transfer (Maturation project TTT P0091) and Lesaffre International (Marcq-en-Barœul, France) and was part of patent n° WO 2020/157297. The authors wish to thank the Sequencing Platform of Toulouse (GeT-Biopuces) and especially Etienne

Rifa for gut microbiota analyses. The authors wish to thank Hervé Robert (Toxalim, Toulouse, France) for helpful discussion.

## Additional information

### Funding
No external funding was received for this work.

### Author contributions
Valérie Laroute, Catherine Beaufrand, Data curation, Formal analysis, Investigation, Methodology; Pedro Gomes, Data curation, Formal analysis, Methodology; Sébastien Nouaille, Valérie Tondereau, Marie-Line Daveran-Mingot, Formal analysis, Investigation, Methodology; Vassilia Theodorou, Conceptualization, Supervision, Validation; Hélène Eutamene, Muriel Mercier-Bonin, Muriel Cocaign-Bousquet, Conceptualization, Funding acquisition, Supervision, Validation, Writing – original draft, Writing – review and editing

### Author ORCIDs
Marie-Line Daveran-Mingot ⬤ http://orcid.org/0000-0001-6884-1840
Hélène Eutamene ⬤ http://orcid.org/0000-0002-2983-1938
Muriel Mercier-Bonin ⬤ http://orcid.org/0000-0001-8398-2529
Muriel Cocaign-Bousquet ⬤ http://orcid.org/0000-0003-2033-9901

### Ethics
Human Subjects: No Animal Subjects: Yes Ethics Statement: Allexperiments were approved by the Local Animal Care and Use Committee (APAFiS#5577-201606061639777v3, see Colorectal and Under PRS conditions sections and APAFiS#14898-2018043016031426, see Without PRS) in compliance with European directive 2010/63/UE.

### Decision letter and Author response
Decision letter https://doi.org/10.7554/eLife.77100.sa1
Author response https://doi.org/10.7554/eLife.77100.sa2

## Additional files

### Supplementary files
• Transparent reporting form

• Supplementary file 1. Specific glutamate decarboxylase activity (µmol/min.mg) of the two *Lactococcus lactis* strains after 7 or 24 hr of growth in M17 supplemented with glutamate (8 g/L), arginine (5 g/L), and NaCl (300 mM; n=3 for each of the two cultures replicates in bioreactor).

• Supplementary file 2. *In vitro* kinetics of γ-aminobutyric acid (GABA) production by *Lactococcus lactis* NCDO2118 and *L. lactis* NCDO2727. GABA production rates (µmol/min) were estimated when bacteria or control vehicle, were in presence of 0.2% (w/v) glutamate at 37°C and pH=4.4, either in 100 mM acetate buffer or with gastric content of naive rat. Two independent replicates were performed.

• Supplementary file 3. Non-parametric comparison of taxa relative abundancies between conditions. Highlighted lines show conditions with significant differences in relative abundancies after Kruskal-Wallis test (p<0.07). Purple highlight indicates that the taxon is more abundant in the faecal microbiota of vehicle-treated animals, green highlight indicates that the taxon is more abundant in the faecal microbiota of NCDO2118-treated animals.

• Supplementary file 4. List of primers. Primers used for the inactivation of gadB in *Lactococcus lactis* NCDO2118.

## Data availability

Relevant additional data files are provided as source data files. For figures 1A 1B and S3: Excel file. For figures 2A 2B, 3A 3B, 4, S1 and S2: txt format. For figure 5: All data (raw and treated) can be found in this link: https://forgemia.inra.fr/umrf/exploremetabar.

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
