## [Editor Report]

Gut microbes produce neuromodulators including the neurotransmitter γ-aminobutyric acid (GABA), which could regulate pain and other neurological outcomes. This study examines how oral administration of the probiotic Lactococcus lactis affects visceral pain in rats, finding that a specific strain of L. lactis (NCD02118) suppresses stress induced pain in a manner dependent on bacterial GadB, the enzyme mediates GABA synthesis.

---

## [Decision Letter]

**Decision letter after peer review:**

[Editors’ note: the authors submitted for reconsideration following the decision after peer review. What follows is the decision letter after the first round of review.]

Thank you for submitting your work entitled "GABA delivery by Lactococcus lactis counteracts stress-induced gut hypersensitivity via GABAB receptor activation in rat" for consideration by *eLife*. Your article has been reviewed by two peer reviewers, and the evaluation has been overseen by a Senior Editor. The reviewers opted to remain anonymous.

Our decision has been reached after consultation between the reviewers. Based on these discussions and the individual reviews below, we regret to inform you that your work will not be considered further for publication in *eLife*.

Overall there was appreciation for your study of the neuromodulatory effects of Lactococcus lactis. However, both reviewers thought additional experiments were required to raise the manuscript to *eLife* standards. A significantly revised manuscript that addresses the reviewers comments below and further dissects the mechanisms by which L. lactis affects the neuronal functions could be of interest to a broad audience.

*Reviewer #1:*

The manuscript is original, well discussed and it touches a topic poorly explored so far which is the neuromodulatory effects of L.lactis and other lactic acid bacteria *in vivo*. However, as it stands, the work will need additional results to be suitable for publication by *eLife* (considering the journal standards). There are already reports on the neuronal effects of GABA secreted by acid lactic bacteria (such as *Lactobacillus* and Streptococci) and Bifidobacterium. The novelty of this present study would be to dissect the mechanisms by which luminal bacteria would affect neuronal functions in the GI tract.

1) The results of GABA measurements in the fecal content clearly show that feeding glutamate alone results in GABA increase in the feces. It is possible that lactic bacteria in the gut microbiota play a role in GABA production. L.lactis NCDO2118, but not L.lactis NCDO2727, may have an impact in gut microbiota composition changing the abundance of bacteria that help the conversion of glutamate into GABA. This distinctive effect of NCDO2118 strain may be related to its high intracellular GAD activity (as compared to NCDO2727) and the role of high amounts of GABA (produced by this strain) in the expansion of other GABA-producing strains in microbiota. This would be compatible with the result obtained with the deficient-GABAB mutant strain of L.lactis NCDO2118. The role of microbiota in the effects reported by the manuscript needs to be investigated.

2) There is no data in the manuscript on the mechanisms by which GABA exerted its visceral anti-hypersensitivity effect during IBS model. Does bacteria-derived GABA access the enteric nervous system (ENS)? How GABA crosses the epithelial barrier? Is GABA binding to GABARB receptors in the neurons or the epithelia (neuroendocrine cells)? GABARB receptors have been identified in both myenteric and submucosal neurons. In the rat mucosal epithelium, positive cells for GABARB have been also detected by immunohistochemistry studies along the length of the GI tract (from the gastric mucosa to the colon), but it has been proposed that the effects of GABA in these receptors differ since neuroendocrine cells are also producers of somatotastin in the gastric epithelium, but GABAR+ positive cells in other parts of the GI tract (such as the duodenum) produced serotonin as well. The novelty of the study would be to shed some light on the mechanisms behind the antinociceptive effects of GABA secreted by luminal L. lactis.

*Reviewer #2:*

The aim of this manuscript is to demonstrate that a strain delivering GABA (L. lactis NCDO2118) in the gut is able exert to anti-nociceptive properties in a stress-induced visceral hypersensitivity rat model. This study is well-conducted, based on strong *in vivo* evidences. They particularly managed to established that the GAD activity from L. lactis and the subsequent GABAergic signaling is a key mechanism in the control of visceral hypersensitivity induced by a stress. Nevertheless, I would have 2 majors comments to make (more detailed above) about (1) the impact on anxiety-like behaviors induced by the stress and (2) the bacteria culture conditions and bacteria growth.

(1) The authors present in their manuscript behavior defects associated with abdominal pains and IBS, is it possible to mimic such symptoms (anxiety, depression,…) using their stress model and if yes, does the NCDO2118 strain can reverse such model induced behavior defects in rats?

(2) The authors present data demonstrating that the NCDO2118 strain produced more GABA in certain culture conditions (pH 4,6) in comparison to the NCDO2727 strain (Figure 1A), but my concern is about the biomass corresponding to the bacterial growth. In fact, when we look at the Figure 1B, we can see that only the NCDO2118 strain present an increase of the biomass, which is almost not the case for the NCDO2727 strain. The authors should comment on this. Does such growth defect could induce a bias in the *in vivo* observed data using such bacteria culture? In addition, the authors should also give similar data about the NCDO2118ΔgadB strain, just to make sure that such deletion did not induce bacterial growth defect as the NCDO2727 strain. Finally, about the *in vivo* experiments, I would be interested to see what would be the results on colonic sensitivity using killed bacteria (heat killed for example).

[Editors’ note: further revisions were suggested prior to acceptance, as described below.]

Thank you for resubmitting your work entitled "*Lactococcus lactis* NCDO2118 exerts visceral antinociceptive properties in rat: increase in GABA concentration and activation of GABAB receptor in the gastro-intestinal tract" for further consideration by *eLife*. Your revised article has been evaluated by Gisela Storz (Senior Editor) and a Reviewing Editor.

The manuscript has been improved but there are some remaining issues that need to be addressed, as outlined below:

Essential revisions:

1) Incorporate additional controls as outlined in detail in the individual reports are critical.

2) Discuss the significance/lack of significance of data.

3) Identify which cell types are involved in the beneficial effect on visceral hypersensitivity.

4) Determine why only 2118 but not 2727 other natural L. lactis strains can produce GABA.

*Reviewer #1*

Laroute et al. investigated the potential anti-hypersensitive properties of a Generally Recognized As Safe (GRAS) lactic acid bacterium, the Lactococcus lactis food-borne bacterium. They have first demonstrated that, among the L. lactis bacteria, only some of them are able to produce active GABA, even if the encoding genes are present in the genome. In addition, those GABA-producing bacteria are able to relieve stress-induced visceral hypersensitivity. Finally, this anti-hypersensitive effect is dependent on the bacteria-produced GABA since genetic and pharmacological inhibition of the GABA leads to a loss of its ability to relieve the stress-induced visceral hypersensitivity.

In general, the presented data are really convincing, and I would need some more controls, in particular in *in vivo* experiments:

1. The *in vitro* data presented in figure 1A nicely demonstrate the difference in the GABA production between 2 strains of L. lactis (NCDO2118 and NCDO2727). I just have a concern about the figure 1B since it seems the non GABA-producing bacteria (NCD02727) present a defect in its growth in comparison to the GABA-producing L. lactis (NCDO2118). May it be a reason why the NCDO2727 did not produce GABA (different growth stage in the bacteria culture) and consequently it loses its *in vivo* properties on visceral hypersensitivity?

2. The *in vivo* results are beautiful and really convincing to me (Figures 2 and 3). I just have the feeling some controls are missing to reinforce the results. On figure 2, I would like to see the effect of the NCDO2118 strain + glutamate in control animals to make sure it will not reduce the basal visceral hypersensitivity? On the figure 3, what happened in PRS + Vehicle + inhibitor, to make sure that the inhibitor does not affect sensitivity by itself?

Finally, I feel that the weakness of the study is in the last figures 4 and 5, since, in my point of view, it did not present as strong data as on the figures 1-3. On the Figure 4, I would appreciate if the authors could increase the n per group since we ca not really see significant differences for the GABA concentration along the GI tract, and it should be more detailed in the discussion. On the figure 5 about microbiota analysis, the authors discussed some differences that are not "significant" as for Escherichia genus since others are not discussed (despite the significance).

*Reviewer #2:*

Lactococcus lactis is a probiotic bacteria that has shown to have multiple beneficial effects on the gut, including anti-inflammatory effects in models of inflammatory bowel diseases. Irritable bowel syndrome (IBS) is characterized by visceral pain. It would be important to define ways to treat IBS and other gut-related disease using gut microbes. The authors have previously found that the L. lactis strain NCDO02118 could synthesize GABA, depending on culture conditions (Laroute et al., Microorganisms 2021; Laroute et al., Front. Microbiol 2016), but the effects *in vivo* in animal models of visceral pain had not been well studied. This study tested two strains of Lactococcus lactis for their ability to produce GABA, and if oral gavage of the strains could modulate visceral hypersensitivity in rats. The authors showed that the strain NCDO02118 could produce GABA *in vitro* and reduce visceral hypersensitivity (measured by colorectal distension) induce by restraint stress. The other strain NCDO02727 did not produce GABA *in vitro* and had no effect on visceral hypersensitivity. The antinociceptive effect from NCDO02118 depended on expression of GadB. Furthermore, the blockade of visceral hypersensitivity could be blocked with a GABAB receptor antagonist, indicating that this receptor mediated pain blockade. GABA levels were also measured in different segments of the gut following oral administration. NCDO02118 administration did not have major impacts on fecal microbiome composition. This study provides evidence that a probiotic delivered into GI tract can alleviate visceral pain and that this could be related to production of GABA.

Strengths:

This study is one of the first to show that deficiency in glutamate dehydroxylase (GAD) component (GadB) in a bacterial strain regulates GABA production and that this has a physiological effect on visceral pain phenotypes in rats. The result that the NCDO02118 strain has antinociceptive properties that are dependent on the GABAB receptors are interesting and has therapeutic implications for IBS and other diseases.

Weaknesses:

The authors suggest that the NCDO02118 strain produces GABA *in vivo*, however, the data from this experiment was not significantly different when GABA levels were measured. The host mechanisms involved blockade of pain *in vivo* are also unclear. For example, the cell types that express GABA receptors can be better characterized and whether GABA signals in specific ways to block the perception of pain. This study could also be strengthened by characterizing why only 2118 but not 2727 or other natural L. lactis strains that can produce GABA, although both of them carry genes that are responsible for GABA metabolism.

Recommendations for the authors:

1. The authors state the NCDO2118 strain increased GABA *in vivo*, however the data in Figure 4 shows a significant decrease of GABA in the cecum and no significant differences along the rest of the GI tract in NCD02118+glutamate vs. vehicle groups (Figure 4A-C). The authors also state that the NDCO2118 strain has a higher concentration of GABA in the stomach in Figure 4C, but this is not a significant difference either. How does this lack of differences correlate with visceral pain blockade? Is it due to poor colonization of NCDO02118 after gavage or improper microenvironment?

2. The authors observed higher GABA in the stomach compared with cecum (Figure 4A). However, the colorectal distension assay was performed in the colon and rectum. How does local GABA upregulation in stomach contribute to the alleviation of visceral pain in the colon? Are circulating GABA levels changed?

3. The authors show that GABAB receptor antagonist blocks the beneficial effect of the NCDO02118 strain on visceral hypersensitivity, but the authors do not try to identify which cell types are involved in this mechanism. Enteric, vagal, and spinal neurons that innervate the gut all have GABA receptors and identifying which subtype is involved would strengthen the study.

4. The authors show an interesting phenotype that the NCDO02727 strain does not produce GABA even though it has the proper machinery. This study does not explore why this strain is not able to make GABA. Can the authors reveal some mechanistic insight into the differences between NCDO02727 vs. NCDO2118?

5. Do gut anti-inflammatory properties of NCDO02118 contribute to their antinociceptive effect in response to PRS? NCDO02118 admininstration induces the upregulation of IL-10 (DOI: 10.3389/fmicb.2021.623920), which can block pain. Is IL-10 also induced in these mice?

6. The authors suggest that L. lactis may be a treatment option for visceral pain and anxiety associated with irritable bowel syndrome (IBS). However, the study does not test whether L. lactis has any effect on anxiety behavior. Can the authors test anxiety behaviors?

---

## [Author Response]

[Editors’ note: the authors resubmitted a revised version of the paper for consideration. What follows is the authors’ response to the first round of review.]

Reviewer #1:The manuscript is original, well discussed and it touches a topic poorly explored so far which is the neuromodulatory effects of L.lactis and other lactic acid bacteria in vivo. However, as it stands, the work will need additional results to be suitable for publication by eLife (considering the journal standards). There are already reports on the neuronal effects of GABA secreted by acid lactic bacteria (such as Lactobacillus and Streptococci) and Bifidobacterium. The novelty of this present study would be to dissect the mechanisms by which luminal bacteria would affect neuronal functions in the GI tract.1) The results of GABA measurements in the fecal content clearly show that feeding glutamate alone results in GABA increase in the feces. It is possible that lactic bacteria in the gut microbiota play a role in GABA production. L.lactis NCDO2118, but not L.lactis NCDO2727, may have an impact in gut microbiota composition changing the abundance of bacteria that help the conversion of glutamate into GABA. This distinctive effect of NCDO2118 strain may be related to its high intracellular GAD activity (as compared to NCDO2727) and the role of high amounts of GABA (produced by this strain) in the expansion of other GABA-producing strains in microbiota. This would be compatible with the result obtained with the deficient-GABAB mutant strain of L.lactis NCDO2118. The role of microbiota in the effects reported by the manuscript needs to be investigated.

We have performed faecal microbiota analyses after L. lactis NCDO2118 treatment and its control using 16S rRNA gene sequencing. The microbiota composition was not globally impacted by the L. lactis NCDO2118 treatment. No changes in the and diversities were observed but few changes at the genus level were detected. These results allowed us to give some additional hypothesis on the potential mechanisms involved, which are discussed in the new version of the manuscript.

2) There is no data in the manuscript on the mechanisms by which GABA exerted its visceral anti-hypersensitivity effect during IBS model. Does bacteria-derived GABA access the enteric nervous system (ENS)? How GABA crosses the epithelial barrier? Is GABA binding to GABARB receptors in the neurons or the epithelia (neuroendocrine cells)? GABARB receptors have been identified in both myenteric and submucosal neurons. In the rat mucosal epithelium, positive cells for GABARB have been also detected by immunohistochemistry studies along the length of the GI tract (from the gastric mucosa to the colon), but it has been proposed that the effects of GABA in these receptors differ since neuroendocrine cells are also producers of somatotastin in the gastric epithelium, but GABAR+ positive cells in other parts of the GI tract (such as the duodenum) produced serotonin as well. The novelty of the study would be to shed some light on the mechanisms behind the antinociceptive effects of GABA secreted by luminal L. lactis.

Concerning the GABA delivery by L. lactis NCDO2118, we measured the GABA level in different compartments of the upper and lower gastrointestinal tract and observed an increase in GABA level in the stomach. Based on this very interesting observation, we introduced in the discussion a new concept of a dynamic ‘virtuous circle’ between L. lactis and its host as described below.

In rodents, previous studies established the presence of GABAB receptors on the mucosal gland cells of the stomach (Castelli et al. 1999; Erdo and Bowery 1986; Erdo et al. 1990; Krantis et al. 1994), which when stimulated increase gastric acid secretion both through vagal cholinergic and gastrin-dependent mechanisms (Piqueras et al. 2004). These combined central and peripheral mechanisms result in an increase in the luminal acidity, which provides a supportive environment for the GAD activity of L. lactis NCDO2118.

Reviewer #2:The aim of this manuscript is to demonstrate that a strain delivering GABA (L. lactis NCDO2118) in the gut is able exert to anti-nociceptive properties in a stress-induced visceral hypersensitivity rat model. This study is well-conducted, based on strong in vivo evidences. They particularly managed to established that the GAD activity from L. lactis and the subsequent GABAergic signaling is a key mechanism in the control of visceral hypersensitivity induced by a stress. Nevertheless, I would have 2 comments to make (more detailed above) about (1) the impact on anxiety-like behaviors induced by the stress and (2) the bacteria culture conditions and bacteria growth.(1) The authors present in their manuscript behavior defects associated with abdominal pains and IBS, is it possible to mimic such symptoms (anxiety, depression,…) using their stress model and if yes, does the NCDO2118 strain can reverse such model induced behavior defects in rats?(2) The authors present data demonstrating that the NCDO2118 strain produced more GABA in certain culture conditions (pH 4,6) in comparison to the NCDO2727 strain (Figure 1A), but my concern is about the biomass corresponding to the bacterial growth. In fact, when we look at the Figure 1B, we can see that only the NCDO2118 strain present an increase of the biomass, which is almost not the case for the NCDO2727 strain. The authors should comment on this. Does such growth defect could induce a bias in the in vivo observed data using such bacteria culture? In addition, the authors should also give similar data about the NCDO2118ΔgadB strain, just to make sure that such deletion did not induce bacterial growth defect as the NCDO2727 strain. Finally, about the in vivo experiments, I would be interested to see what would be the results on colonic sensitivity using killed bacteria (heat killed for example).

Whatever the bacterial strain tested and considering the growth profile differences, we systematically adjusted the cell amount in order to give in 1 mL by oral route the same CFU to the rats (10exp9 CFU/day/rat). The text of the manuscript has been revised accordingly to avoid any misunderstanding. Furthermore, the mutation of the gadB gene in L. lactis NCDO2118 has no impact on the cell growth. This is now added in the manuscript (supplementary data Figure S2).

In terms of basic knowledge, our input will undoubtedly open new perspectives for GRAS L. lactis strains as future therapeutic agents for the management of visceral pain and the anxious profile of IBS patients. We consider the most relevant audience for this work is large and multidisciplinary, including researchers in Nutrition, Microbiology, Digestive pathoph ysiology but also clinicians, gastroenterologists and industrial partners working on functional food (probiotics and postbiotics). General public could also be interested for human and animal well-being.

[Editors’ note: what follows is the authors’ response to the second round of review.]

Essential revisions:1) Incorporate additional controls as outlined in detail in the individual reports are critical.

We have now incorporated additional controls (reviewer 1 point 2). Indeed, we have verified that *L. lactis* NCDO2118 in presence of glutamate had no impact on basal CRD sensitivity. We have now added a dedicated figure (Figure 2—figure supplement 1) on the supplementary information section. Comments have been included in the main text of the revised manuscript L164-L166. Regarding the effect of the inhibitor on sensitivity by itself, a previous *in vivo* study (dorsal spinal cord microdialysis) aimed at evaluating the role of GABAB receptor antagonism (using SCH-50911, the same as in our study) on GAT-1 inhibition-induced effects on evoked algesic amino acid (ASP, GLU, GlY) release in the dialysates. This study reported that SCH-50911 administered alone did not affect the evoked release of those amino acids (Smith *et al.* 2007). Furthermore, according to the 3R rules, our objective was to reduce at the maximum the number of animals used in our *in vivo* studies. For all these reasons, we did not test a possible effect *per se* of SCH-50911 in our model.

2) Discuss the significance/lack of significance of data.

In response to this point (reviewer 1 point 3), we have performed additional experiments on *in vitro* kinetics of GABA production by *L. lactis* NCDO2118 and *L. lactis* NCDO2727 in the presence of glutamate under “stomach-like” conditions (at pH=4.6 in acetate buffer or gastric juice sampled from naive rats) (now presented in Supplementary File 2). We clearly demonstrate that, notably in gastric juice sampled from naive rats and supplemented with glutamate, NCDO2118 strain produced GABA. We found that the rate of GABA production by the NCDO2727 strain was similar to the vehicle but the rate of GABA production by the NCDO2118 strain was fourfold increased. To reinforce our findings on the potential role of the gastric region, we have inverted both sets of data in the revised version of the manuscript and slightly modified the *in vitro* part. Comments have also been added in the Discussion section L351-L354.

3) Identify which cell types are involved in the beneficial effect on visceral hypersensitivity.

As answered to the reviewer (reviewer 2 point 3), in this study we did not identify the cell types involved in the GABA-dependent- antinociceptive properties of NCDO2118. In the gastro-intestinal tract, GABA receptors are present on the vagal afferent fibers both at the mucosal endings that respond to touch and chemical stimuli and muscular endings that respond optimally to mechanical stretch or tension (Page *et al.* 2002). As discussed in our manuscript, GABAB receptors are also widely distributed from the stomach to the ileum in the enteric nervous system (ENS). In rodents, the presence of GABAB receptors has also been identified on the mucosal gland cells of the stomach. Taken together, all these data suggest complex mechanisms and pathways by which GABA delivered by NCDO2118 strain exerts visceral antinociceptive effect. Based on the literature and our own observations, at this step of our knowledge, we cannot emphasize a cell type more than another and additional investigations are needed to open this black box.

4) Determine why only 2118 but not 2727 other natural L. lactis strains can produce GABA.

We have shown in the manuscript (results L153-L156 and Supplementary File 1) the very strong differences in the intracellular GAD activity in the two strains NCDO2118 and NCDO2727 explaining their major GABA production differences (and also their different antinociceptive properties).

In the manuscript, we have demonstrated by *in silico* analyses that these differences were not due to differences in the genetic organisation of *gadCB* operon and *gadR* gene nor to differences in GadB,C and R protein sequences (L140-L144).

To determine whether these differences in GABA production between strains might be associated to differences in the regulation of *gadCB* and *gadR* expression at the transcription level, we have now analysed the nucleotide sequences of their promoters. They were found to be almost identical and the very small number of mutations identified (2 single nucleotide polymorphisms for *gadR* and 1 for *gadCB*) seems unlikely to cause large differences in transcriptional regulation (this is described in more details in the comment 4 for the reviewer 2).

We believe that the differences between NCDO2218 and NCDO2727 strains are likely due to posttranscriptional regulations but we do not really know what are the mechanisms involved. Nothing in the literature can help us to formulate realistic hypothesis.

As suggested by the reviewer 1, the growth defect of the NCDO2727 strain could participate to the lack of efficient GABA production. To check the relevance of this mechanism, we have investigated the growth and the GABA production in another culture medium enhancing the growth performances of the NCDO2727 strain. We demonstrated that the GABA production and the growth rate are not correlated (see below in the response to comments 1 and 2 of reviewer 1).

In order to study the *gadCB* regulation and thanks to reporter gene constructions, we are currently investigating the *gadCB* expression in various environments and in various strains. Further work will be required to fully understand *gad* operon regulations and explain the differences observed between the strains.

In the manuscript, in the results and in the Discussion sections, we have added sentences on the promoter similarities (see the comment 4 of the reviewer 2) and also on the lack of relationship between GABA production and bacterial cell growth (see the comment 1 of the reviewer 1).

Reviewer #1Laroute et al. investigated the potential anti-hypersensitive properties of a Generally Recognized As Safe (GRAS) lactic acid bacterium, the Lactococcus lactis food-borne bacterium. They have first demonstrated that, among the L. lactis bacteria, only some of them are able to produce active GABA, even if the encoding genes are present in the genome. In addition, those GABA-producing bacteria are able to relieve stress-induced visceral hypersensitivity. Finally, this anti-hypersensitive effect is dependent on the bacteria-produced GABA since genetic and pharmacological inhibition of the GABA leads to a loss of its ability to relieve the stress-induced visceral hypersensitivity.In general, the presented data are really convincing, and I would need some more controls, in particular in in vivo experiments:1. The in vitro data presented in figure 1A nicely demonstrate the difference in the GABA production between 2 strains of L. lactis (NCDO2118 and NCDO2727). I just have a concern about the figure 1B since it seems the non GABA-producing bacteria (NCD02727) present a defect in its growth in comparison to the GABA-producing L. lactis (NCDO2118). May it be a reason why the NCDO2727 did not produce GABA (different growth stage in the bacteria culture) and consequently it loses its in vivo properties on visceral hypersensitivity?

Indeed, the NCDO2727 strain grows slowly in the medium used in the experiments depicted in our manuscript (i.e. M17 supplemented with glutamate). Since then, we have tested another culture medium, YE medium with glutamate. The cell growth was significantly enhanced (growth rate of 0.7 h^-1^ compared to 0.24 h^-1^ in the condition described in the manuscript) but the GABA production remains very low (less than 0.3 mM, which is of the same order of magnitude as the level described here). This result indicates the growth limitation of the strain NCDO2727 is not the reason of its lack of GABA production (and the loss of antinociceptive effect).

In the manuscript, we have added a sentence in the Results section L147: “The GABA concentration did not increase when the biomass production increased during growth of NCDO2727 strain in YE medium with glutamate (data not shown)” and a comment in the discussion L304-L305 “The growth defect of NCDO2727 should also not be involved since higher growth of the strain in another culture medium did not restore a high GABA production.”

2. The in vivo results are beautiful and really convincing to me (Figures 2 and 3). I just have the feeling some controls are missing to reinforce the results. On figure 2, I would like to see the effect of the NCDO2118 strain + glutamate in control animals to make sure it will not reduce the basal visceral hypersensitivity? On the figure 3, what happened in PRS + Vehicle + inhibitor, to make sure that the inhibitor does not affect sensitivity by itself?

We have verified that *L. lactis* NCDO2118 in presence of glutamate had no impact on basal CRD sensitivity. For clarity reasons, we have added a dedicated figure (Figure 2—figure supplement 1) on the supplementary information section rather than inserting data on Figure 2. Comments have been included in the main text of the revised manuscript L164-L166. Regarding the comment of the reviewer on the effect of the inhibitor on sensitivity by itself, a previous *in vivo* study (dorsal spinal cord microdialysis) aimed at evaluating the role of GABAB receptor antagonism (using SCH-50911, the same as in our study) on GAT-1 inhibition-induced effects on evoked algesic amino acid (ASP, GLU, GlY) release in the dialysates. This study reported that SCH-50911 administered alone did not affect the evoked release of those amino acids (Smith *et al.* 2007). Furthermore, according to the 3R rules, our objective was to reduce at the maximum the number of animals used in our *in vivo* studies. For all these reasons, we did not test a possible effect *per se* of SCH-50911 in our model.

– Smith CG, Bowery NG, Whitehead KJ. GABA transporter type 1 (GAT-1) uptake inhibition reduces stimulated aspartate and glutamate release in the dorsal spinal cord *in vivo* via different GABAergic mechanisms. Neuropharmacology. 2007 Dec;53(8):975-81. doi: 10.1016/j.neuropharm.2007.09.008. Epub 2007 Sep 29. PMID: 17981306.

Finally, I feel that the weakness of the study is in the last figures 4 and 5, since, in my point of view, it did not present as strong data as on the figures 1-3. On the Figure 4, I would appreciate if the authors could increase the n per group since we ca not really see significant differences for the GABA concentration along the GI tract, and it should be more detailed in the discussion. On the figure 5 about microbiota analysis, the authors discussed some differences that are not "significant" as for Escherichia genus since others are not discussed (despite the significance).

Our objective was to determine in which GIT compartment GABA delivery specifically occurred for NCDO2118 strain (compared to vehicle and NCDO2727) and even though results did not reach significance, we can see in the stomach that the GABA production by *L. lactis* NCDO2118 is higher than that measured in vehicle or NCDO2727 strain. Interestingly, in a follow-up study (article in preparation) considering a higher GABA-producing *L. lactis* strain than NCDO2118, the GABA concentration in the stomach was significantly higher than for NCDO2727 strain and nearly for vehicle (p=0.07).

We have performed additional experiments presented in Supplementary File 2 on *in vitro* kinetics of GABA production by *L. lactis* NCDO2118 and *L. lactis* NCDO2727 in the presence of glutamate under “stomach-like” conditions (at pH=4.6 in acetate buffer or gastric juice sampled from naive rats). We clearly demonstrate that, notably in gastric juice sampled from naive rats and supplemented with glutamate, NCDO2118 strain produced GABA. In particular, we found that the rate of GABA production by the NCDO2727 strain was similar to the vehicle but the rate of GABA production by the NCDO2118 strain was fourfold increased. To reinforce our findings on the potential role of the gastric region, we have inverted both sets of data in the revised version of the manuscript and slightly modified the *in vitro* part. Comments have been added in the Discussion section L351-L354.

Regarding the impact on the gut microbiota, we fully agree with the reviewer that we observed a significantly increased abundance in *Frisingicoccus* and *Papillibacter* genera (p-value of 0.041 for both), complementarily to *Clostridium* (p-value of 0.023) and *Escherichia* (p-value of 0.069). According to the reviewer’s comment, we have added these two genera in the description of the results on the gut microbiota, while introducing some slight changes. In particular, the p-values have been added. However, in the discussion part, we still focus on (i) *Clostridium* genus since it was shown *in silico* as modulating GABA (i.e. GABA producer and consumer) in humans and (ii) on *Escherichia* genus since such enrichment in the faecal microbiota of *L. lactis*-treated animals could help reduce visceral pain via a lipopeptide-mediated GABA functionalization pathway. To the best of our knowledge, nothing is known on the *Frisingicoccus* and *Papillibacter* genera regarding GABA.

Reviewer #2:[…]Recommendations for the authors:1. The authors state the NCDO2118 strain increased GABA in vivo, however the data in Figure 4 shows a significant decrease of GABA in the cecum and no significant differences along the rest of the GI tract in NCD02118+glutamate vs. vehicle groups (Figure 4A-C). The authors also state that the NDCO2118 strain has a higher concentration of GABA in the stomach in Figure 4C, but this is not a significant difference either. How does this lack of differences correlate with visceral pain blockade? Is it due to poor colonization of NCDO02118 after gavage or improper microenvironment?

As indicated above (see our response to comment 3 of reviewer 1), our objective was to determine in which GIT compartment GABA delivery specifically occurred for NCDO2118 strain (compared to vehicle and NCDO2727) and even though results did not reach significance, we can see in the stomach that the GABA production by *L. lactis* NCDO2118 is higher than that measured in vehicle or NCDO2727 strain. Interestingly, in a follow-up study (article in preparation) considering a higher GABA-producing *L. lactis* strain than NCDO2118, the GABA concentration in the stomach was significantly higher than for NCDO2727 strain and nearly for vehicle (p=0.07). We have performed additional experiments presented in Supplementary File 2 on *in vitro* kinetics of GABA production by *L. lactis* NCDO2118 and *L. lactis* NCDO2727 in the presence of glutamate under “stomach-like” conditions (at pH=4.6 in acetate buffer or gastric juice sampled from naive rats). We clearly demonstrate that, notably in gastric juice sampled from naive rats and supplemented with glutamate, the NCDO2118 strain produced a fourfold-increased GABA level while the rate of GABA production by the NCDO2727 strain was similar to the vehicle. To reinforce our findings on the potential role of the gastric region, we have inverted both sets of data in the revised version of the manuscript and slightly modified the *in vitro* part. Comments have been added in the Discussion section L353-L354. Concerning the detection of *L. lactis* in the GIT environment (from stomach to feces), a PCR-based approach would be insightful to overcome the poor suitability of the 16S rRNA gene sequencing method to low abundant bacterial populations. Interestingly, in the study of Duranti *et al.* in *Scientific Reports* in 2020 (https://doi.org/10.1038/s41598-020-70986-z) on *Bifidobacterium adolescentis*, the authors showed that the fecal concentration of GABA in rats treated with two high GABA-producing strains, namely *B. adolescentis* PRL2019 and *B. adolescentis* HD17T2H, was not statistically different than that found in rats treated with a non-GABA producer strain B. adolescentis ATCC15703 or rats not supplemented by *B. adolescentis* strains (control group). Furthermore, in the study of Pokusaeva *et al.* in *Neurogastroenterology and Motility* in 2017 (https://doi.org/10.1111/nmo.12904), demonstrating that GABA-producing *Bifidobacterium dentium* modulates visceral sensitivity in the intestine, the authors found that cecal GABA concentrations in the *B. dentium*-treated mice vs PBS-treated ones were within a similar range (10.1 ± 2.5 vs 10.3 ± 1.7 μg/g cecal content). This clearly indicates that it is not so straightforward to correlate cecal/fecal levels of GABA and *in vivo* efficacy.

2. The authors observed higher GABA in the stomach compared with cecum (Figure 4A). However, the colorectal distension assay was performed in the colon and rectum. How does local GABA upregulation in stomach contribute to the alleviation of visceral pain in the colon? Are circulating GABA levels changed?

We thank the reviewer for addressing these questions. Reflex among different organs along the gastrointestinal tract is well known. For example, ingestion of food into the stomach induces a response not only in the stomach but also in the small intestine and colon; distention of the rectum inhibits gastric and intestinal motility (Bampton *et al.* 2002; Kerlin *et al.* 1983; Shafik *et al.* 2000). Cross-talk along the gastrointestinal tract has also been observed. Gastric electrical stimulation inhibits rectal tone, an effect mediated by sympathetic pathway (Liu *et al.* 2005).Vagal afferent fibers innervating the upper gastro-intestinal tract play a critical role in the initiation of symptoms and reflexes controlling several functions (Page and Blackshaw, 1999). Vagal afferents are comprised of mucosal endings that respond to touch and chemical stimuli and muscular endings that respond optimally to mechanical stretch or tension (Page *et al.* 2002). There is a clear distribution of GABA receptors and particularly of GABAB receptors along the peripheral afferent vagal fibers. In our study, even though we have to more deeply elucidate in a follow-up study the functionalization pathway of GABA released by *L. lactis* into the stomach, we can hypothesize a primary mucosal activation of such receptors with central vagal pathways in the nucleus tractus solitarii (NTS) and dorsal vagal nucleus activations. In response to this central GABAergic integrative stimulation, an analgesic spinal descending outcome could in turn counteract visceral hypersensitivity induced by stress in response to colorectal distension. Besides this nervous cross-talk mechanistic pathway of GABA delivered by *L. lactis* into the stomach, we cannot exclude an antinociceptive effect mediated by an increase of circulating GABA. However, at this stage we have no data on this hypothesis. Thanks to the referee’s input, the measurements of circulating GABA levels are now planned in our future investigations on *L. lactis* studies.

– Bampton PA, Dinning PG, Kennedy ML, Lubowski DZ, and Cook IJ. The proximal colonic motor response to rectal mechanical and chemical stimulation. Am J Physiol Gastrointest Liver Physiol 282: G443–G449, 2002. doi: 10.1152/ajpgi.00194.2001. PMID: 11841994.

– Kerlin P, Zinsmeister A, and Phillips S. Motor responses to food of the ileum, proximal colon, and distal colon of healthy humans. Gastroenterology 84: 762–770, 1983. PMID: 6825988.

– Shafik A and El-Sibai O. Esophageal and gastric motile response to rectal distension with identification of a recto-esophago gastric reflex. Int J Surg Investig 1: 373–379, 2000. PMID: 11341593.

– Liu S, Wang L, Chen J. D. Z. Cross-talk along gastrointestinal tract during electrical stimulation: effectsand mechanisms of gastric/colonic stimulation on rectal tone in dogs. Am J Physiol Gastrointest Liver Physiol 288: G1195–G1198, 2005. doi: 10.1152/ajpgi.00554.2004. Epub 2005 Feb 3. PMID: 15691864.

– Page AJ and Blackshaw LA. GABA(B) receptors inhibit mechanosensitivity of primary afferent endings. J Neurosci19: 8597–8602, 1999. doi: 10.1523/JNEUROSCI.19-19-08597.1999. PMID: 10493759; PMCID: PMC6783028.

– Page AJ, Martin CM and Blackshaw LA (2002). Vagal mechanoreceptors and chemoreceptors in mouse stomach and esophagus. J Neurophysiol 87, 2095–2103. doi: 10.1152/jn.00785.2001. PMID: 11929927.

3. The authors show that GABAB receptor antagonist blocks the beneficial effect of the NCDO02118 strain on visceral hypersensitivity, but the authors do not try to identify which cell types are involved in this mechanism. Enteric, vagal, and spinal neurons that innervate the gut all have GABA receptors and identifying which subtype is involved would strengthen the study.

We agree with the reviewer that in this study we did not identify the cell types involved in the antinociceptive mechanism. As answered in the comment 2 of the reviewer, GABA receptors are present on the vagal afferent fibers both at the mucosal endings that respond to touch and chemical stimuli and muscular endings that respond optimally to mechanical stretch or tension (Page *et al.* 2002, see above). As discussed in our paper, GABAB receptors are also widely distributed from the stomach to the ileum in the enteric nervous system (ENS). In rodents, the presence of GABAB receptors has also been identified on the mucosal gland cells of the stomach. All these data suggest complex mechanisms and pathways by which GABA delivered by NCDO2118 strain exerts visceral antinociceptive effect. Based on the literature and our own observations, at this step of our knowledge, we cannot emphasize a cell type more than another and we agree that additional investigations are needed to open this black box.

4. The authors show an interesting phenotype that the NCDO02727 strain does not produce GABA even though it has the proper machinery. This study does not explore why this strain is not able to make GABA. Can the authors reveal some mechanistic insight into the differences between NCDO02727 vs. NCDO2118?

Why the NCDO2118 strain produces GABA and the corresponding GAD enzyme but not the NCDO2727 is a really intriguing question. In the manuscript, we have demonstrated by *in silico* analyses that these differences were not due to differences in the genetic organisation of *gadCB* operon and *gadR* gene nor to differences in GadB,C and R protein sequences.

In order to determine if these differences in GABA production could be associated to differences in *gadCB* and *gadR* expression regulation, we have now analysed the nucleotide sequences of their promoters. They were found identical (except 2 SNPs for *gadR* and 1 for *gadCB*, see Author response image 1 the example regarding the comparison of the *gadB* promoter in NCDO2118 and NCDO2727). These very few mutations seem minimal to explain major differences in transcriptional regulation of GadB.

**Author response image 1. sa2fig1:** 

We believe that the differences between NCDO2118 and NCDO2727 strains are likely due to post-transcriptional regulations but we do not really know the precise mechanisms involved. In the literature, very few studies on the regulation of the *gadCB/R* gene have been reported, even for chloride activation, for which molecular basis is still unclear (Sanders *et al.* 1998). In fact, nothing in the literature can helps us to formulate realistic hypothesis.In order to study the *gadCB* and *gadR* regulation and thanks to reporter gene constructions, we are currently investigating the *gadCB* expression in various environments and in various strains. Considerable work will be required to fully understand the *gad* operon regulations and identify the involved actors (protein, ncRNA, etc.). Only after understanding this, we can expect to explain the differences observed between the strains.

At this stage, we added in the manuscript the results of the promoter analyses. The following sentence was added in results L142-L144 “Only two single nucleotide polymorphisms and a deletion of one bp were identified in the *gadR* and *gadCB* promoters respectively compared to NCDO2118.” In the discussion a comment was also added L302-L304 “Because the promoters of *gad* genes are nearly identical in the two strains, transcriptional regulations should not explain the observed differences.”

– Sanders JW, Leenhouts K, Burghoorn J, Brands JR, Venema G, Kok J. A chloride-inducible acid resistance mechanism in Lactococcus lactis and its regulation. Mol Microbiol. 1998 Jan;27(2):299-310. doi: 10.1046/j.1365-2958.1998.00676.x. PMID: 9484886.

5. Do gut anti-inflammatory properties of NCDO02118 contribute to their antinociceptive effect in response to PRS? NCDO02118 admininstration induces the upregulation of IL-10 (DOI: 10.3389/fmicb.2021.623920), which can block pain. Is IL-10 also induced in these mice?

We thank the reviewer for this relevant question. In a previous study, we have shown that PRS increased gut paracellular permeability and endotoxemia, two responses which in cascade induced elevated mRNA expression of pro-inflammatory cytokines (IL-1β, IL-6 and TNF-α) in the PVN (Ait-Belgnaoui *et al.* 2012, see above). This central neuro-inflammation was prevented by a chronic daily treatment with a probiotic strain i.e. *Lactobacillus farciminis*. Since several probiotics have been described to be able to restore gut physical barrier damage as well as to increase anti-inflammatory cytokines like IL-10, it would be very interesting in future investigations to evaluate gut anti-inflammatory properties of NCDO2118 in this model.

6. The authors suggest that L. lactis may be a treatment option for visceral pain and anxiety associated with irritable bowel syndrome (IBS). However, the study does not test whether L. lactis has any effect on anxiety behavior. Can the authors test anxiety behaviors?

We thank the reviewer for this point. We know that the comorbidity between IBS and anxiety disorders is elevated and that anxiety and depression contribute to inefficient therapies in IBS (Vandvik *et al.* 2006; Popa *et al.* 2015). Anxiety is a complex disorder, defined as the response to prolonged, unpredictable threat, a response which encompasses physiological, affective, and cognitive changes. To our knowledge, different animal models are developed to mimic anxiety. The Open Field Maze is one of the most commonly anxiety-like models used to characterize behaviors in rodents. However, until now, we haven’t this type of model in our lab but we are exploring the possibility of a collaborative work to investigate *L. lactis* ability to reduce anxiety behavior in response to stress.

Besides, even if we did not use an anxiety-like model in our study, we kindly point that PRS model is a reliable model widely used for studying the pharmacological modulation of the GABAergic system in order to prevent the anxiety-like behavior induced by acute and chronic stress (Reznikov *et al.* 2009; Nuss, 2015; Assad *et al.* 2020; Farajdokht *et al.* 2020).

– Assad, N., Luz, W. L., Santos-Silva, M., Carvalho, T., Moraes, S., Picanço-Diniz, D., et al. (2020). Acute restraint stress evokes anxiety-like behavior mediated by telencephalic inactivation and gabaergic dysfunction in zebrafish brains. Sci. Rep. 10:5551. doi: 10.1038/s41598-020-62077-w. PMID: 32218457; PMCID: PMC7099036.

– Farajdokht, F., Vosoughi, A., Ziaee, M., Araj-Khodaei, M., Mahmoudi, J., and Sadigh-Eteghad, S. (2020). The role of hippocampal GABAAreceptors on anxiolytic effects of Echium amoenum extract in a mice model of restraint stress. Mol. Biol. Rep. 47, 6487–6496. doi: 10.1007/s11033-020-05699-7. Epub 2020 Aug 10. PMID: 32778988.

– Nuss, P. (2015). Anxiety disorders and GABA neurotransmission: a disturbance of modulation. Neuropsychiatr. Dis. Treat. 11, 165–175. doi: 10.2147/NDT.S58841. PMID: 25653526; PMCID: PMC4303399.

– Popa Stefan-Lucian, Dumitrascu Dan Lucian. Anxiety and IBS revisited: ten years later. Clujul Med. 2015;88(3):253-7. doi: 10.15386/cjmed-495. Epub 2015 Jul 1. PMID: 26609253; PMCID: PMC4632879.

– Reznikov Leah R, Reagan Lawrence P, Fadel Jim R. Effects of acute and repeated restraint stress on GABA efflux in the rat basolateral and central amygdala. Brain Res. 2009 23;1256:61-8. doi: 10.1016/j.brainres.2008.12.022. Epub 2008 Dec 24. PMID: 19124010.

– Vandvik PO, Lydersen S, Farup PG. Prevalence, comorbidity and impact of irritable bowel syndrome in Norway. Scand J Gastroenterol. 2006;41(6):650–656. doi: 10.1080/00365520500442542. PMID: 16716962.